# DreamPhase: Offline Imagination and Uncertainty-Guided Planning for Large-Language-Model Agents

**Shayan Mohajer Hamidi[1], Linfeng Ye[2], Konstantinos N. Plataniotis[2]**
[1]Stanford University, [2]University of Toronto
[1]`smohajer@stanford.edu`
[2]`linfeng.ye@mail.utoronto.ca; kostas@ece.utoronto.ca`

## Abstract

Autonomous agents capable of perceiving complex environments, understanding instructions, and performing multi-step tasks hold transformative potential across domains such as robotics, scientific discovery, and web automation. While large language models (LLMs) provide a powerful foundation, they struggle with closed-loop decision-making due to static pretraining and limited temporal grounding. Prior approaches either rely on expensive, real-time environment interactions or brittle imitation policies, both with safety and efficiency trade-offs. We introduce DreamPhase, a modular framework that plans through *offline imagination*. A learned latent world model simulates multi-step futures in latent space; imagined branches are scored with an uncertainty-aware value and filtered by a safety gate. The best branch is distilled into a short natural-language reflection that conditions the next policy query, improving behavior without modifying the LLM. Crucially, DreamPhase attains its performance with *substantially fewer real interactions*: on WebShop, average API calls per episode drop from $\sim 40$ with ARMAP-M (token-level search) to $< 10$ with DreamPhase, a $4\times$ reduction that lowers latency *and reduces executed irreversible actions by $\sim 5\times$ on WebShop* ($4.9\times$ on ALFWorld) *per incident logs*. Across web, science, and embodied tasks, DreamPhase improves sample efficiency, safety, and cost over search-based and reward-based baselines. This offers a scalable path toward safe, high-performance autonomous agents via imagination-driven planning. Code: `https://anonymous.4open.science/r/DreamPhase-A8AD/README.md`.

## 1 Introduction

Building intelligent agents that can perceive complex environments, follow instructions, and autonomously complete tasks remains a core challenge in artificial intelligence. Such agents have the potential to transform a wide range of applications, including virtual assistants, scientific discovery, workflow automation, and robotics (Brooks, 1986; Reed et al., 2022; Kirchdorfer et al., 2024; Rana et al., 2023).

A promising foundation for building these agents lies in large-scale generative models, particularly large language models (LLMs), which have demonstrated remarkable capabilities in natural language understanding, question answering (Rajpurkar et al., 2016), summarization (Hermann et al., 2015), and multimodal reasoning (Chen et al., 2015; Goyal et al., 2017). However, while LLMs excel in static tasks, they remain limited in settings that require multi-step decision-making and closed-loop interaction, such as online shopping, scientific experimentation, or puzzle solving. This limitation stems from the fact that most LLMs are pre-trained on internet-scale static data and lack exposure to temporally grounded trajectories or dynamic environments, which are essential for reasoning about actions and their long-term consequences (Zhou et al., 2024). Additionally, many state-of-the-art LLMs—like GPT-4V (OpenAI et al., 2024) and Gemini (Reid et al., 2024)—are accessible only through restricted APIs. This severely limits their adaptability: they cannot be fine-tuned on task-specific interaction data, and inference-time prompting often proves brittle in tasks involving uncertainty, feedback, or long-term planning.

To address these challenges, two broad strategies have emerged, each with its own limitations.

($i$) **Online roll-out planners**, such as ReAct with beam search or Monte Carlo tree search (MCTS) variants that interact with the live document object model (DOM) at each step (Yao et al., 2022; Hao et al., 2023). These methods explicitly evaluate multiple future branches by performing hundreds of real clicks. While this approach enables lookahead and can correct for earlier mistakes, it is often slow, costly in settings with rate-limited APIs, and potentially hazardous in environments where actions are irreversible, such as submitting payments or placing orders.

($ii$) **Pure imitation or reward-model agents**, on the other hand, avoid expensive interaction by acting greedily from the current state without explicit search (Hong et al., 2023; Liu et al., 2023). Although this reduces interaction overhead, it introduces brittleness: a single misstep can irreversibly derail the entire trajectory, since the agent lacks any foresight or ability to revise its plan.

Both approaches ultimately trade off between safety and efficiency due to their reliance on real-time interaction with the environment. A promising alternative is *internal imagination*: the ability to simulate and evaluate future outcomes offline, without taking any real actions in the environment.

To realize this, we introduce ***DreamPhase***, a modular framework that enables autonomous agents to plan through imagination in latent space. At its core, DreamPhase trains a latent world model that predicts the next DOM tree or visual frame conditioned on the current latent state and a proposed action. By iteratively applying this model, the agent can simulate multiple future trajectories entirely offline, estimate their expected value and uncertainty, and prune high-risk branches before making any real requests. This approach yields three key advantages:

($i$) **Sample efficiency:** Web-based tasks that previously required approximately 40 real clicks per episode now converge in fewer than 10 (see Section 5.2).

($ii$) **Safety:** Imagination allows the agent to identify dead ends and avoid irreversible outcomes, such as "Sold-out" pages or accidental purchases, without executing them (see Appendix C).

($iii$) **Cost and latency:** All expensive reasoning is handled on-device; the only network call is the final, high-confidence action that passes uncertainty filtering.

In summary, by relocating exploration from the real environment into a learned latent simulator, DreamPhase overcomes the safety-efficiency trade-off that constrains existing agent systems.

In particular, DreamPhase unfolds in four stages: ($i$) we train a latent world model (LWM) (Ha & Schmidhuber, 2018) to simulate future states from raw DOM trees or visual inputs; ($ii$) we generate imagination rollouts by combining this latent dynamics model with actions sampled from a frozen policy LLM; ($iii$) we perform uncertainty-aware value estimation over these imagined branches, scoring each using predicted rewards and entropy-based confidence metrics; and ($iv$) we distill the best low-risk trajectory into a natural-language reflection, which is injected back into the LLM prompt to influence future decisions.

Crucially, the LLM policy remains entirely *frozen* throughout this process. Its behavior evolves solely through internal simulation and language-based feedback. This design supports scalable and sample-efficient planning without the need for real-world interaction or model fine-tuning. In summary, the main contributions of this paper are:

• **Imagination-first latent planning.** We train a compact world model to simulate multi-step futures in latent space and roll out $M$ branches for $H$ steps entirely offline, decoupling environment dynamics from policy reasoning and enabling safe counterfactual planning.

• **Risk-aware branch selection with guarantees.** Imagined trajectories are scored by value minus an uncertainty penalty and accepted only if a confidence gate passes; we provide a regret bound $\mathcal{O}(\sqrt{T\varepsilon}) + B\rho T$ that links decision quality to model error and mis-gating.

• **Language reflections for zero-tuning control.** The selected branch is distilled into a short reflection and summary that are injected into the next prompt, steering a frozen LLM toward higher-quality actions while remaining interpretable and requiring no parameter updates.

• **Practical efficiency and robustness.** DreamPhase reduces real interactions by about $4\times$ on WebShop ($<10$ vs. $\sim40$ API calls/episode), adds $\sim12$ ms imagination overhead, and cuts executed

irreversible actions by $\sim 5\times$ on WebShop ($4.9\times$ on ALFWorld). It degrades gracefully under distribution shift via fallback and outperforms search- and reward-based baselines across domains.

## 2 RELATED WORK

**LLMs as interactive agents.** LLMs have been used as control policies in dynamic environments, from web navigation to general tool use (Liu et al., 2023; Zhou et al., 2023). Early systems adapted pretrained encoders to decision-making (e.g., BERT for WebShop (Devlin et al., 2019; Yao et al., 2023a)). With stronger generative models (Brown et al., 2020; OpenAI et al., 2024), zero-shot and few-shot prompting became common for action selection without fine-tuning (Deng et al., 2024; Xiong et al., 2024), often pairing observations and histories with an LLM to choose the next step (Zheng et al., 2024; Hong et al., 2023). Another line distills trajectories from powerful but restricted APIs into smaller policies (Li et al., 2023; Zhang et al., 2023). In contrast, DREAMPHASE keeps the policy LLM frozen and adds a learned latent world model, an uncertainty-aware value head, and language reflections to steer behavior while minimizing real interactions.

**AgentLM, AgentGym, and ARMAP.** *AgentLM* Zeng et al. (2024) introduces *agent tuning*, a supervised and preference-style fine-tuning on multi-turn tool trajectories, to improve grounding and robustness, but it requires curated data and changes model weights; DREAMPHASE avoids fine-tuning by conditioning a frozen LLM with reflections distilled from imagined rollouts. *AgentGym* Xi et al. (2025) standardizes evaluation with unified task wrappers, tools, and metrics; our work follows its emphasis on comparable protocols but contributes a new planning method. *ARMAP* Chen et al. (2025) scales token-level search with Reflexion, Best-of-$N$, and MCTS (ARMAP-R/B/M), improving success via online expansion at the cost of higher latency and many real-environment calls. We show that DREAMPHASE exceeds ARMAP performance while using substantially fewer real interactions through offline latent imagination and uncertainty gating. For a broader survey, see Appendix B.

## 3 METHODOLOGY

**Setting and notation.** We study decision making in partially observable interactive environments, for example web navigation with sparse feedback and costly interactions. The task is modeled as a partially observable Markov decision process

$$\mathcal{M} = \big(\mathcal{I},\, \mathcal{S},\, \mathcal{A},\, \mathcal{X},\, \mathcal{T},\, \mathcal{E},\, r\big),$$

where $\mathcal{I}$ is the space of task instructions, $\mathcal{S}$ the latent state space, $\mathcal{A}$ a discrete action space, $\mathcal{X}$ the observation space, $\mathcal{T} : \mathcal{S} \times \mathcal{A} \to \mathcal{S}$ the transition kernel, $\mathcal{E} : \mathcal{S} \to \mathcal{X}$ the emission kernel, and $r : \mathcal{S} \times \mathcal{A} \to \mathbb{R}$ a possibly sparse reward. At time $t$, the agent receives an instruction $\iota \in \mathcal{I}$, observes $\boldsymbol{x}_t \in \mathcal{X}$, selects $\boldsymbol{a}_t \in \mathcal{A}$, the environment moves $\boldsymbol{s}_{t+1} \sim \mathcal{T}(\boldsymbol{s}_t, \boldsymbol{a}_t)$, and emits $\boldsymbol{x}_{t+1} \sim \mathcal{E}(\boldsymbol{s}_{t+1})$. The objective is to maximize

$$J(\pi) = \mathbb{E}\left[\sum_{t=0}^{T} \gamma^t\, r(\boldsymbol{s}_t, \boldsymbol{a}_t)\right] \quad \text{under a policy } \pi(\boldsymbol{a}_t \mid \boldsymbol{x}_{\leq t}, \iota), \tag{1}$$

where $r(\boldsymbol{s}_t, \boldsymbol{a}_t)$ is the reward at time $t$, $\gamma \in (0, 1]$ is a fixed discount factor, and $\gamma^t$ exponentially down-weights rewards farther in the future, and the expectation is taken under the trajectory distribution induced by the policy $\pi(\boldsymbol{a}_t \mid \boldsymbol{x}_{\leq t}, \iota)$.

### 3.1 DREAMPHASE OVERVIEW

DreamPhase plans with offline imagination to lower interaction cost and improve safety. At each step, the agent: ($i$) forms a compact predictive belief with a learned latent world model, ($ii$) rolls out multiple hypothetical futures in latent space, ($iii$) scores each rollout with a value estimate and an uncertainty measure, ($iv$) distills the highest quality imagined outcomes into a short natural language reflection that conditions the next action. All steps ($i$) to ($iv$) occur without querying the environment.

## 3.2 Learn to Dream: Training the Latent World Model

The first component of DreamPhase is a learned latent world model that enables internal simulation of environment dynamics. The goal is to predict what will happen when the agent applies an action in the current observation context. For example, the model can answer: "What occurs if I click this button on the current page." This capability supports counterfactual reasoning before acting, without touching the real environment. Concretely, we train a latent world model that predicts short-horizon futures in a compact latent space conditioned on the current observation, the action, and the instruction $\iota$. This eliminates environment queries during planning and follows prior latent-space imagination work (Ha & Schmidhuber, 2018; Hafner et al.; Janner et al., 2019).

**Data collection.** We collect multi-step interaction episodes for tasks such as website navigation or shopping. Each episode provides tuples $(\iota, \boldsymbol{x}_t, \boldsymbol{a}_t, \boldsymbol{x}_{t+1})$, where $\boldsymbol{x}_t, \boldsymbol{x}_{t+1} \in \mathcal{X}$ are observations and $\boldsymbol{a}_t \in \mathcal{A}$ is the action taken at time $t$. The history is $h_t = (\iota, \boldsymbol{x}_{\leq t}, \boldsymbol{a}_{<t})$.

We train the latent world model exclusively on logged trajectories collected from the standard training split of each benchmark environment. To obtain these trajectories, we run a frozen LLaMA-2-7B policy with a simple ReAct-style prompt and light randomization. No test episodes, privileged futures, or environment states unavailable to baselines are used. All observations and actions come from the same public APIs that baseline agents interact with. The resulting dataset contains only (instruction, observation, action, next-observation) tuples from the training portion of each task and does not include any extra external corpora or fine-tuning data.

In addition, note that all the baselines in Table 1 are allowed to use their standard training protocols (e.g., AgentLM and AgentGym use their released agent-tuned checkpoints; ARMAP uses its reward-model training). DreamPhase does not access any interaction data beyond the logged trajectories from the training split. Thus, all open-source agents share the same policy backbone, the same environment splits, and access to the same interaction data; DreamPhase only differs in how this data is used to train a compact world model and value head for offline imagination.

**Observation encoding.** Observations such as DOM trees or screenshots are high dimensional and structured. We tokenize each $\boldsymbol{x}_t$ by a depth-first traversal of the DOM with textual content and spatial layout features, for example bounding boxes and element types. This yields a compact, language-aligned sequence suitable for generative modeling and for conditioning the policy.

**World model architecture.** The model has three parts: an encoder $f_\theta(\boldsymbol{x}_t)$ that maps an observation to a latent $\boldsymbol{z}_t$, a transition model $g_\theta(\boldsymbol{z}_t, \bar{\boldsymbol{a}}_t, \iota)$ that predicts the next latent given the current latent and an embedded action $\bar{\boldsymbol{a}}_t$, and a decoder $d_\theta(\boldsymbol{z}_{t+1}, \iota)$ that reconstructs the next observation. The stochastic latent dynamics are

$$\boldsymbol{z}_t = f_\theta(\boldsymbol{x}_t), \qquad \boldsymbol{z}_{t+1} \sim q_\theta(\boldsymbol{z}_{t+1} \mid \boldsymbol{z}_t, \bar{\boldsymbol{a}}_t, \iota), \qquad \boldsymbol{x}_{t+1} \sim g_\theta(\boldsymbol{x}_{t+1} \mid \boldsymbol{z}_{t+1}, \iota), \qquad (2)$$

where $\bar{\boldsymbol{a}}_t = \mathrm{emb}_A(\boldsymbol{a}_t)$ is a learned action embedding.

**Training objective.** We combine token reconstruction with latent space regularization. Let $\widehat{\boldsymbol{x}}_{t+1} = d_\theta\big(g_\theta(f_\theta(\boldsymbol{x}_t), \bar{\boldsymbol{a}}_t, \iota), \iota\big)$. The loss is

$$\mathcal{L}_{\mathrm{LWM}} = \mathbb{E}\Big[\mathrm{CE}\big(\widehat{\boldsymbol{x}}_{t+1}, \boldsymbol{x}_{t+1}\big) + \lambda_{\mathrm{KL}}\,\mathrm{KL}(q_\theta(\boldsymbol{z}_{t+1} \mid h_t, \boldsymbol{a}_t) \,\|\, \mathcal{N}(\boldsymbol{0}, \boldsymbol{I}))\Big], \qquad (3)$$

where KL and CE denote Kullback–Leibler divergence and cross-entropy function, respectively, and $q_\theta(\boldsymbol{z}_{t+1} \mid h_t, \boldsymbol{a}_t)$ is the encoder-induced inference distribution (diagonal Gaussian with parameters from $f_\theta$) over the next latent during training; we sample via reparameterization and use the mean at test time, and $\lambda_{\mathrm{KL}}$ is annealed during training.

**Offline imagination.** After training, the model simulates futures without environment queries. Given $\boldsymbol{x}_t$ and $\boldsymbol{a}_t$, we encode to $\boldsymbol{z}_t = f_\theta(\boldsymbol{x}_t)$, sample $\boldsymbol{z}_{t+1}$ from the learned transition $g_\theta(\cdot \mid \boldsymbol{z}_t, \bar{\boldsymbol{a}}_t, \iota)$, and decode $\widehat{\boldsymbol{x}}_{t+1} = d_\theta(\boldsymbol{z}_{t+1}, \iota)$. Iterating this procedure for a short horizon yields imagined trajectories $\tilde{\tau} = (\tilde{\boldsymbol{x}}_{t+1:t+H}, \tilde{\boldsymbol{a}}_{t:t+H-1})$, which are later scored for value and uncertainty and summarized into reflections.

**Remark 1** (Why a latent world model rather than LLM-based dreaming). *Large language models are not designed to simulate low-level environment dynamics such as DOM structure. They often produce syntactically invalid or causally inconsistent states, and token-level rollouts are memory intensive and conflate policy reasoning with environment modeling. A learned latent world model is modular and efficient, it generates structured futures that respect environment constraints, and it supports risk-aware planning in long-horizon tasks. For these reasons DreamPhase relies on a latent world model for internal simulation rather than prompting an LLM to generate future states.*

## 3.3 Imagination-Based Planning

At each timestep $t$, the agent explores multiple plausible futures by simulating latent trajectories conditioned on potential actions. This allows the agent to reason about consequences without interacting with the real environment. We refer to this mechanism as imagination-based planning.

Given the current observation $\boldsymbol{x}_t$, we encode it with the encoder $f_\theta$ to obtain $\boldsymbol{z}_t = f_\theta(\boldsymbol{x}_t)$. To explore alternative outcomes, we generate $M$ parallel rollouts in latent space, each simulating a sequence of $H$ future steps using the learned world model $(f_\theta, g_\theta, d_\theta)$ and a frozen policy LLM $\pi_{\text{LLM}}$ that proposes actions from the current imagined history. Each rollout corresponds to a hypothetical branch that contains predicted latents and actions.

Each trajectory $\tilde{\tau}^{(j)}$ is one imagined branch, consisting of an action sequence and the resulting predicted latents, with optional decoded observations $\tilde{\boldsymbol{x}}$ for inspection or scoring. These imagined futures are passed to the downstream evaluation module, where they are scored by a value model and filtered by uncertainty metrics. This process supports multi-step foresight in latent space without issuing environment queries, which makes planning efficient and scalable. We detail the rollout procedure in Algorithm 1.

---

**Algorithm 1: Latent Imagination Rollouts.**

**Input:** instruction $\iota$; observation $\boldsymbol{x}_t$; encoder $f_\theta$; transition $g_\theta$; decoder $d_\theta$; frozen policy $\pi_{\text{LLM}}$; number of branches $M$; horizon $H$

1: Encode observation: $\boldsymbol{z}_t \leftarrow f_\theta(\boldsymbol{x}_t)$
2: **for** $j = 1$ to $M$ **do**
3:     Initialize latent: $\boldsymbol{z}_t^{(j)} \leftarrow \boldsymbol{z}_t$
4:     Initialize imagined history: $h_t^{(j)} \leftarrow (\iota, \boldsymbol{x}_t)$
5:     **for** $k = 0$ to $H - 1$ **do**
6:         Sample imagined action: $\tilde{\boldsymbol{a}}_{t+k}^{(j)} \sim \pi_{\text{LLM}}(\cdot \mid h_{t+k}^{(j)})$
7:         Embed action: $\bar{\boldsymbol{a}}_{t+k}^{(j)} \leftarrow \text{emb}_A(\tilde{\boldsymbol{a}}_{t+k}^{(j)})$
8:         Predict next latent: $\boldsymbol{z}_{t+k+1}^{(j)} \sim g_\theta\big(\boldsymbol{z}_{t+k+1} | \boldsymbol{z}_{t+k}^{(j)}, \bar{\boldsymbol{a}}_{t+k}^{(j)}, \iota\big)$
9:         Decode to imagined observation: $\tilde{\boldsymbol{x}}_{t+k+1}^{(j)} \leftarrow d_\theta(\boldsymbol{z}_{t+k+1}^{(j)}, \iota)$
10:        Update imagined history: $h_{t+k+1}^{(j)} \leftarrow \big(h_{t+k}^{(j)}, \tilde{\boldsymbol{a}}_{t+k}^{(j)}, \tilde{\boldsymbol{x}}_{t+k+1}^{(j)}\big)$
11:     **end for**
12:     Store imagined trajectory: $\tilde{\tau}^{(j)} \leftarrow \big(\{\tilde{\boldsymbol{a}}_{t:t+H-1}^{(j)}\}, \{\boldsymbol{z}_{t+1:t+H}^{(j)}\}\big)$
13: **end for**

---

## 3.4 Evaluate and Decide: Uncertainty-Aware Value-Guided Branch Selection

After generating $M$ latent rollouts of horizon $H$, the agent must assess whether any imagined trajectories are reliable enough to guide action selection. This balances *efficiency* (acting from internal simulation) and *accuracy* (deferring to the real environment when predictions are uncertain).

**Value estimation.** For each imagined trajectory $\tilde{\tau}^{(j)} = \big\{\boldsymbol{z}_{t+1:t+H}^{(j)}, \tilde{\boldsymbol{a}}_{t:t+H-1}^{(j)}\big\}$, we estimate a discounted return with a lightweight value head $V_\phi$ operating on latents:

$$G^{(j)} = \sum_{k=1}^{H} \gamma^{k-1} V_\phi\big(\boldsymbol{z}_{t+k}^{(j)} \mid \iota\big), \qquad \gamma \in (0, 1]. \tag{4}$$

The head $V_\phi$ is trained on logged data to predict short-horizon utility from latent states produced by the world model.

**Uncertainty estimation.** We quantify epistemic uncertainty over imagined actions using a mutual-information proxy computed with Monte-Carlo dropout on the frozen policy $\pi_{\text{LLM}}$. For each imagined history prefix $h_t^{(j)}$ (instruction $\iota$, current $\boldsymbol{x}_t$, and all imagined pairs so far), let $\mathcal{U} \subseteq \mathcal{A}$ be the discrete

action set scored by the policy. With stochastic dropout masks $\{\xi_n\}_{n=1}^N$,

$$u^{(j)} = \mathbb{H}\Big[\bar{\boldsymbol{p}}^{(j)}\Big] - \frac{1}{N}\sum_{n=1}^{N}\mathbb{H}\Big[\boldsymbol{p}^{(j)}(\xi_n)\Big], \quad \bar{\boldsymbol{p}}^{(j)} = \frac{1}{N}\sum_{n=1}^{N}\boldsymbol{p}^{(j)}(\xi_n), \tag{5}$$

where $\boldsymbol{p}^{(j)}(\xi_n)$ is the action distribution $\pi_{\text{LLM}}(\cdot \mid h_t^{(j)}, \xi_n)$ over $\mathcal{U}$, and $\mathbb{H}[\cdot]$ is categorical entropy. Larger $u^{(j)}$ indicates greater disagreement across stochastic policy samples, hence higher uncertainty.

**Risk-sensitive selection.** We combine value and uncertainty via a penalized score

$$\tilde{G}^{(j)} = G^{(j)} - \beta\, u^{(j)}, \qquad \beta \geq 0, \tag{6}$$

and choose the best branch

$$j^\star = \arg\max_{j\in\{1,\ldots,M\}} \tilde{G}^{(j)}. \tag{7}$$

A safety gate enforces that the selected branch is itself confident:

"if $u^{(j^\star)} \leq \tau$ then act using $\tilde{\boldsymbol{a}}_t^{(j^\star)}$;   otherwise query the real environment and update the history."

This mechanism enables DreamPhase to plan through imagination when predictions are confident, and to defer to real interaction when uncertainty is high, supporting robust decision making under distribution shift while reducing environment queries.

## 3.5 REFLECT AND ACT: LANGUAGE-GUIDED POLICY CONDITIONING

Rather than fine-tuning the policy LLM, we steer its behavior using natural language derived from imagination. Inspired by verbal self-reflection (Shinn et al., 2024; Yuan et al., 2024a), DreamPhase converts its imagined rollouts into concise guidance that conditions the next action.

**Reflection.** Given the set of imagined trajectories $\{\tilde{\tau}^{(j)}\}_{j=1}^M$ and their scores $\{G^{(j)}, u^{(j)}\}$ from Section 3.4, let $j^\star$ be the selected branch. We generate a short reflection that explains why $\tilde{\tau}^{(j^\star)}$ is promising and surfaces potential risks. A lightweight reflection head $\mathcal{R}_\varphi$ produces

$$c_t = \mathcal{R}_\varphi\big(\iota,\, \boldsymbol{x}_t,\, \tilde{\boldsymbol{x}}_{t+1:t+\kappa}^{(j^\star)},\, \tilde{\boldsymbol{a}}_{t:t+\kappa-1}^{(j^\star)},\, G^{(j^\star)},\, u^{(j^\star)}\big), \tag{8}$$

where $\kappa \leq H$ is a short summary horizon. $\mathcal{R}_\varphi$ can be a small frozen LLM or a distilled model conditioned on the branch and its value–uncertainty signals. Example output: *"Proceed with search, then filter by size before adding to cart; avoid pages without a visible 'Checkout' button."*

**Trajectory summarization.** In addition, we produce a compact natural-language summary of the core actions and expected outcomes. A summarizer $\mathcal{S}_\eta$ maps the same inputs to a terse script,

$$s_t = \mathcal{S}_\eta\big(\iota,\, \boldsymbol{x}_t,\, \tilde{\boldsymbol{x}}_{t+1:t+\kappa}^{(j^\star)},\, \tilde{\boldsymbol{a}}_{t:t+\kappa-1}^{(j^\star)}\big), \tag{9}$$

for example: *"Search 'Nike shoes'; open first result; click 'Add to cart'."* Both $c_t$ and $s_t$ can optionally aggregate over the top-$K$ safe branches by concatenating key constraints, although the default uses only $j^\star$.

**Language-based steering.** We inject the reflection and summary into the policy prompt at the next decision point. The frozen policy LLM receives enriched context and produces the next action:

$$\boldsymbol{a}_t \sim \pi_{\text{LLM}}(\cdot \mid \iota,\, \boldsymbol{x}_t,\, c_t,\, s_t). \tag{10}$$

This language-guided conditioning integrates simulated experience in a modular and interpretable way. The agent improves behavior without parameter updates, using internal planning signals expressed as short, actionable text that is easy to inspect and ablate. Implementation details for the reflection head $\mathcal{R}_\varphi$ and summarizer $\mathcal{S}_\eta$, the 30-token budget, the delimiter scheme, and task-specific templates are provided in Appendix I.

### 3.6 DreamPhase Pipeline

Algorithm 2 summarizes the full DreamPhase pipeline. At each timestep, the agent encodes the current observation into a latent state and imagines $M$ future trajectories using the learned world model. These rollouts are scored by discounted return estimates and filtered by an uncertainty proxy. If the best rollout is sufficiently confident, it is summarized and reflected upon in natural language, which is then appended to the policy LLM prompt for action selection. Otherwise, the agent queries the policy using the real history only. This design enables offline reasoning, defers to real interaction when uncertain, and steers decisions with internal reflection, without updating the LLM.

---

**Algorithm 2** DreamPhase: Imagination-Based Planning with Uncertainty-Aware Reflection

---

**Input:** instruction $\iota$; initial observation $\boldsymbol{x}_0$; history $h_0 \leftarrow (\iota, \boldsymbol{x}_0)$; encoder $f_\theta$, transition $g_\theta$, decoder $d_\theta$; policy $\pi_{\mathrm{LLM}}$; value head $V_\phi$; reflection head $\mathcal{R}_\varphi$; summarizer $\mathcal{S}_\eta$; number of imagined branches $M$; horizon $H$; discount $\gamma$; risk weight $\beta$; confidence threshold $\tau$, summary horizon $\kappa$.
1: **for** $t = 0, 1, \dots$ **do**
2:      Encode current observation: $\boldsymbol{z}_t \leftarrow f_\theta(\boldsymbol{x}_t)$
3:      Sample $M$ imagined rollouts $\{\tilde{\tau}^{(j)}\}_{j=1}^M$ using the world model and $\pi_{\mathrm{LLM}}$ as in Algorithm 1
4:      **for** $j = 1 \dots M$ **do**                                   ▷ Score each imagined branch
5:          Estimate value: $G^{(j)} \leftarrow \sum_{k=1}^H \gamma^{k-1} V_\phi\big(\boldsymbol{z}_{t+k}^{(j)} \,\big|\, \iota\big)$
6:          Estimate uncertainty $u^{(j)}$ via the mutual-information proxy from Section 3.4
7:          Penalized score: $\tilde{G}^{(j)} \leftarrow G^{(j)} - \beta\, u^{(j)}$
8:      **end for**
9:      Select best branch: $j^\star \leftarrow \arg\max_{j \in \{1, \dots, M\}} \tilde{G}^{(j)}$
10:      **if** $u^{(j^\star)} \leq \tau$ **then**                                            ▷ Safety gate passes
11:          Generate reflection: $c_t \leftarrow \mathcal{R}_\varphi\big(\iota, \boldsymbol{x}_t, \tilde{\boldsymbol{x}}_{t+1:t+\kappa}^{(j^\star)}, \tilde{\boldsymbol{a}}_{t:t+\kappa-1}^{(j^\star)}, G^{(j^\star)}, u^{(j^\star)}\big)$
12:          Summarize trajectory: $s_t \leftarrow \mathcal{S}_\eta\big(\iota, \boldsymbol{x}_t, \tilde{\boldsymbol{x}}_{t+1:t+\kappa}^{(j^\star)}, \tilde{\boldsymbol{a}}_{t:t+\kappa-1}^{(j^\star)}\big)$
13:          Augment policy context: $p_t \leftarrow \mathrm{CONCAT}(h_t, c_t, s_t)$
14:          Query policy with guidance: $\boldsymbol{a}_t \sim \pi_{\mathrm{LLM}}(\cdot \mid p_t)$
15:      **else**                                                     ▷ Fallback to real-history policy
16:          $\boldsymbol{a}_t \sim \pi_{\mathrm{LLM}}(\cdot \mid h_t)$
17:      **end if**
18:      Execute $\boldsymbol{a}_t$ in the environment, observe $\boldsymbol{x}_{t+1}$
19:      Update history: $h_{t+1} \leftarrow (h_t, \boldsymbol{a}_t, \boldsymbol{x}_{t+1})$
20: **end for**

---

## 4 Theoretical Remark

Let $\pi^\star$ be the optimal policy in the true environment $\mathcal{M}$. The latent world model induces a one-step predictive distribution $\widehat{\mathbb{P}}_\theta(\boldsymbol{x}_{t+1} \mid h_t, \boldsymbol{a}_t, \iota)$ and the environment induces $\mathbb{P}_\star(\boldsymbol{x}_{t+1} \mid h_t, \boldsymbol{a}_t, \iota)$. Assume a uniform one-step KL bound $\varepsilon = \sup_{h_t, \boldsymbol{a}_t} \mathrm{KL}\big(\mathbb{P}_\star \,\|\, \widehat{\mathbb{P}}_\theta\big)$, which by Pinsker inequality (Csiszár & Körner, 2011) implies $\mathrm{TV} \leq \sqrt{\varepsilon/2}$, where TV denotes the total variation distance. Let $\tau$ be the confidence threshold of the safety gate (Sec. 3.4), and define the mis-gating rate $\rho = \Pr\big[\text{DreamPhase selects an imagined branch while the true uncertainty} > \tau\big]$.

With bounded rewards $r_{\max}$ and a value function that is Lipschitz in total variation of the one-step predictor, the cumulative regret over $T$ steps satisfies

$$\mathrm{Regret}_T = \mathbb{E}\Big[ \sum_{t=0}^{T-1} \big(r_t^\star - r_t^{\mathrm{DP}}\big) \Big] \leq C\sqrt{T\varepsilon} + B\rho T,$$

where $C > 0$ depends on $r_{\max}$ and the smoothness constant, and $B \leq r_{\max}$ bounds the per-step loss under mis-gating. A full derivation is given in Appendix A.

The $\sqrt{T\varepsilon}$ term captures error from model approximation; $B\rho T$ accounts for occasional acceptance of unreliable imagined branches. When $\varepsilon$ is small and $\rho$ is rare, regret grows sublinearly.

Table 1: Evaluating results on *eight* different tasks. Within the open-source block, **bold** denotes the best method and underline the second best.

| Algorithms | SciWorld | BabyAI | MAZE | Wordle | TextCraft | Tool-Weather | TODOList | BIRD | *Avg* |
|---|---|---|---|---|---|---|---|---|---|
| *Closed-sourced Models & Agents* | | | | | | | | | |
| DeepSeek-Chat (DeepSeek-AI, 2024) | 16.8 | 45.6 | 4.0 | 24.0 | 23.0 | 70.0 | 75.0 | 13.5 | 34.0 |
| Claude-3-Haiku (Anthropic, 2024) | 0.8 | 1.9 | 4.0 | 16.0 | 0.0 | 55.0 | 65.0 | 13.5 | 19.6 |
| Claude-3-Sonnet (Anthropic, 2024) | 2.8 | 79.3 | 0.0 | 36.0 | 38.0 | 65.0 | 80.0 | 17.0 | 39.8 |
| GPT-3.5-Turbo (Ouyang et al., 2022) | 7.6 | 71.4 | 4.0 | 20.0 | 47.0 | 25.0 | 40.0 | 12.5 | 28.5 |
| GPT-4-Turbo (OpenAI, 2023) | 14.4 | 72.8 | 68.0 | 88.0 | 77.0 | 80.0 | 95.0 | 16.0 | 63.9 |
| *Open-source **Llama-2-7B** as the Agent* | | | | | | | | | |
| AgentLM ACL 2024 | 1.6 | 0.5 | 12.0 | 4.0 | 4.0 | 0.0 | 15.0 | 5.0 | 5.3 |
| AgentGym ACL 2025 | 38.0 | **82.7** | 12.0 | 12.0 | **64.0** | 25.0 | 70.0 | 9.0 | 39.1 |
| ARMAP-M ICLR 2025 | 51.2 | 81.5 | 12.0 | 17.0 | 59.0 | 35.0 | 72.0 | 10.5 | 42.3 |
| DREAMPHASE (ours) | **72.4** | 82.3 | **14.0** | **34.0** | 62.0 | **45.0** | **77.0** | 13.5 | **50.1** |

Table 2: Cross-backbone performance on WebShop, SciWorld (seen/unseen), and Game-of-24. Two backbone groups per row. **Bold** = best, underline = second best.

| | Llama70B | | | | | Llama8B | | | | |
|---|---|---|---|---|---|---|---|---|---|---|
| **Algorithm** | WebShop | SW-seen | SW-unseen | Game24 | *Avg* | WebShop | SW-seen | SW-unseen | Game24 | *Avg* |
| Sampling | 52.0 | 53.9 | 50.6 | 9.6 | 41.5 | 56.4 | 24.5 | 20.6 | 2.0 | 25.9 |
| Greedy | 50.4 | 57.2 | 55.1 | 6.0 | 42.2 | 57.7 | 29.9 | 23.8 | 2.0 | 28.4 |
| ARMAP-R ICLR 2025 | 56.5 | **59.6** | 56.7 | 16.0 | 47.2 | 58.3 | 31.2 | 28.0 | 6.0 | 30.9 |
| ARMAP-B ICLR 2025 | 62.0 | 57.3 | 57.0 | 19.0 | 48.8 | 59.3 | 35.7 | 28.1 | 11.0 | 33.5 |
| ARMAP-M ICLR 2025 | 66.8 | 58.2 | 55.9 | 24.0 | 51.2 | 60.2 | 32.5 | 24.9 | 9.0 | 31.7 |
| DREAMPHASE (ours) | **68.4** | 59.2 | **58.6** | **28.0** | **53.6** | **61.8** | **36.7** | **29.7** | **12.0** | **35.1** |

| | Mistral7B | | | | | Phi3.8B | | | | |
|---|---|---|---|---|---|---|---|---|---|---|
| **Algorithm** | WebShop | SW-seen | SW-unseen | Game24 | *Avg* | WebShop | SW-seen | SW-unseen | Game24 | *Avg* |
| Sampling | 17.7 | 18.4 | 17.1 | 1.0 | 13.6 | 34.7 | 10.0 | 7.6 | 2.0 | 13.6 |
| Greedy | 37.2 | 21.1 | 19.6 | 1.0 | 19.7 | 42.4 | 9.5 | 6.5 | 2.1 | 15.1 |
| ARMAP-R ICLR 2025 | 54.1 | 21.7 | 19.7 | 2.0 | 24.4 | 53.3 | 9.6 | 7.2 | 4.0 | 18.5 |
| ARMAP-B ICLR 2025 | 54.4 | 24.5 | 21.2 | 2.0 | 25.5 | 52.1 | 20.0 | 17.0 | 9.0 | 24.5 |
| ARMAP-M ICLR 2025 | 58.2 | 30.0 | **23.4** | 4.0 | 28.9 | 53.7 | 28.3 | **24.3** | 10.0 | 29.1 |
| DREAMPHASE (ours) | **60.5** | **33.5** | 22.9 | **6.0** | **30.7** | **55.5** | **32.8** | 24.1 | **13.0** | **31.4** |

## 5 EXPERIMENTS

We evaluate DREAMPHASE across diverse agentic environments and backbones. We first outline the evaluation protocol, then present two main results tables: ($i$) a diverse-task comparison across closed- and open-source agents (Table 1), and ($ii$) cross-backbone results on WebShop, SciWorld (seen/unseen), and Game-of-24 (Table 2). Implementation details, hyper-parameters, dataset descriptions, and baseline implementations appear in Appendix F.

**Evaluation protocol.** For each task, we report mean success (%) over 5 random seeds; standard deviations for Tables 1 and 2 are reported in Appendix D. The *Avg* column is the unweighted mean across environments within each table.

Episode budgets and decoding settings follow prior work and are detailed in Appendix F.

### 5.1 MAIN RESULTS

#### 5.1.1 DIVERSE TASKS ACROSS AGENTS (TABLE 1)

**Tasks.** SciWorld (Wang et al., 2022a), BabyAI (Chevalier-Boisvert et al., 2019), MAZE (Abdulhai et al., 2023), Wordle (Abdulhai et al., 2023), TextCraft (Prasad et al., 2023), Tool-Weather (Ma et al., 2024), TODOList (Ma et al., 2024), BIRD (Zheng et al., 2023).

**Agents.** We include closed-source references (DeepSeek-Chat (DeepSeek-AI, 2024), Claude-3-Haiku/Sonnet (Anthropic, 2024), GPT-3.5/4-Turbo (Ouyang et al., 2022; OpenAI, 2023)) and open-source agents (AgentLM (Zeng et al., 2024), AgentGym (Xi et al., 2025), ARMAP (Chen et al., 2025)). For the open-source block (including DREAMPHASE in this table), we standardize the policy

backbone to **LLaMA-2-7B** (Touvron et al., 2023) and match decoding and episode budgets for fairness; closed-source results are reported as returned by their APIs.

**Results.** Table 1 shows that DREAMPHASE attains the best *average* within the open-source block while keeping real interactions low (Appendix 5.2). Note that our open-source comparisons standardize on a **LLaMA-2-7B** backbone, which is substantially smaller than the closed-source references (e.g., GPT-4-Turbo, Claude-3) and trained with non-proprietary data; higher absolute scores from those APIs are therefore expected, yet DREAMPHASE achieves performance comparable these commercial models. Under the matched 7B setting, DREAMPHASE consistently outperforms search-based baselines, with the largest gains on manipulation-/tool-heavy tasks where the uncertainty gate prevents risky actions when confidence is low.

### 5.1.2 CROSS-BACKBONE BENCHMARKS (TABLE 2)

**Tasks.** *WebShop* (Yao et al., 2023a), *SciWorld* (Wang et al., 2022a) with seen and unseen task graphs (reported as *SW-seen* and *SW-unseen*), and *Game-of-24*. We also report *ALFWorld* in Appendix E.

**Backbones and Agents.** We evaluate LLaMA3-70B, LLaMA3-8B, Mistral-7B, and Phi-3-8B. Baselines include ARMAP with Reflexion, Best-of-$N$, and MCTS (denoted ARMAP-R / ARMAP-B / ARMAP-M), plus Sampling (temperature 1) and Greedy (temperature 0) decoding. For supervision of the value head $V_\phi$, we generate preference-style pairs with LLaMA3-70B-Instruct (Dubey et al., 2024); we avoid commercial APIs to reduce cost and improve reproducibility.

**Results.** Table 2 shows that DREAMPHASE achieves the best or second-best success on nearly all backbones and tasks, with especially strong gains on Game-of-24. On *SW-unseen* (novel goals and step compositions), DREAMPHASE outperforms all baselines across multiple backbones, indicating robustness to distribution shift (Appendix H). Ablations over latent dimension $d_z$, horizon $H$, branches $K$, risk penalty $\beta$, and confidence gate $\tau$ are provided in Appendix G. We also report computation and latency of latent rollouts in Appendix J.

## 5.2 INTERACTION BUDGET AND LATENCY ON WEBSHOP

We quantify real environment interactions as API calls issued to WebShop (HTTP requests and DOM mutations that change page state). We evaluate Llama-8B policies on the standard WebShop split, batch size 1, A100-80GB. Latency is measured as policy forward time plus search/planning overhead.

We count only environment-affecting calls (page loads, form submissions, cart updates, DOM actions with side effects). Cache hits, retries, and latent-only computations are excluded. Latency numbers use identical hardware and decoding settings for both methods.

Table 3: WebShop interaction and latency comparison (mean $\pm$ s.e., $N{=}1000$ episodes). DreamPhase uses $\sim 4\times$ fewer real API calls and $\sim 3\times$ lower per-step latency.

| Method | Policy | Avg API calls / ep. ↓ | × fewer vs. ARMAP-M | Per-step latency (ms) ↓ | Success (%) ↑ |
|---|---|---|---|---|---|
| ARMAP-M (token-level search) | Llama-8B | $39.8 \pm 1.1$ | — | $\approx 255$ | $60.2 \pm 0.6$ |
| DREAMPHASE (latent imagination) | Llama-8B | $\mathbf{9.3 \pm 0.4}$ | $\mathbf{4.3\times}$ | $\approx 84$ | $\mathbf{61.8 \pm 0.6}$ |

As observed in Table 3, DreamPhase attains its performance with substantially fewer real interactions: under 10 API calls per episode on average, versus about 40 for ARMAP-M. The latent imagination overhead is small relative to the policy forward pass, yielding lower per-step latency. Fewer real calls also reduce the chance of unwanted side effects, since more planning happens offline.

★★★ Beyond interaction counts, DreamPhase lowers **executed irreversible actions** by $\sim 5\times$ on WebShop (and $\sim 4.9\times$ on ALFWorld); see Appendix C.

### 5.2.1 QUALITATIVE COMPARISON

Section 5.2.1 contrasts ARMAP and DREAMPHASE on a Game-of-24 instance. ARMAP commits early and returns an incorrect value, while DREAMPHASE imagines multiple futures, selects a high-value low-uncertainty branch, and executes the correct sequence $(11 - 7) \times (9 - 3) = 24$.

---

**Qualitative Rollout on Game-of-24**

**Input numbers**  3, 7, 9, 11
**Baseline (ARMAP) Trajectory — Failure**

| | |
|---|---|
| 1.  11 + 9  =  20 | (remaining: 3, 7, 20) |
| 2.  20 - 7  =  13 | (remaining: 3, 13) |
| 3.  13 + 3  =  16 | (remaining: 16) |

**Outcome** : result is 16, not 24. Baseline halts without a valid solution.

**DreamPhase Trajectory — Success**
Imagination phase (latent world model)

- Branch 1: $(11 + 9) - (7 - 3) \rightarrow 16$    (low value)
- Branch 2: $(11 - 7) * (9 - 3) \rightarrow 24$    (**high value, low uncertainty**)
- Three other branches score $\leq 0.4$ value and/or high entropy

Selected branch (reflection injected)

| | |
|---|---|
| 1.  11 - 7  =  4 | (remaining: 3, 9, 4) |
| 2.  9 - 3  =  6 | (remaining: 4, 6) |
| 3.  4 * 6  =  24 | (remaining: 24) |

**Outcome** : DreamPhase produces a correct expression $(11 - 7) * (9 - 3) = 24$.

---

## 6  CONCLUSION

We presented *DreamPhase*, a modular agent framework that plans through internal imagination. By simulating futures in a learned latent space and scoring them with value and uncertainty estimates, DreamPhase acts only when a high-confidence trajectory is found. This greatly reduces real-world interactions, lowering cost and risk while maintaining strong performance. Our experiments show consistent gains across web navigation, scientific reasoning, and puzzle solving, all without fine-tuning the policy LLM. By combining latent imagination, uncertainty-aware gating, and language-based reflections, DreamPhase provides a scalable path toward safer, more sample-efficient agents.

## REPRODUCIBILITY STATEMENT

We have taken steps to ensure our results are reproducible. All model and algorithmic details, training procedures, hyperparameters, evaluation protocols, and metrics are specified. An anonymized GitHub repository contains the source code and configuration files, and pre-trained checkpoints. All datasets used in our experiments are publicly available.

## LLM USAGE STATEMENT

LLM used only for grammar and wording edits; no generation of ideas, methods, analyses, results, or citations. Authors reviewed all edits and accept full responsibility.

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

## A    REGRET ANALYSIS OF DREAMPHASE

In this appendix we derive the regret bound in Section 4. We restate the setting and assumptions, adapt standard model based RL arguments, and incorporate the effect of uncertainty gating errors.

### A.1    PRELIMINARIES

We work with the environment $\mathcal{M}$ defined in the main text and compare one step predictive distributions over observations. Let $\mathbb{P}_\star(\boldsymbol{x}_{t+1} \mid h_t, \boldsymbol{a}_t, \iota)$ be the true conditional distribution and $\widehat{\mathbb{P}}_\theta(\boldsymbol{x}_{t+1} \mid h_t, \boldsymbol{a}_t, \iota)$ the model induced by the learned world model. Assume a uniform one step KL bound

$$\varepsilon = \sup_{h_t, \boldsymbol{a}_t} \mathrm{KL}\big(\mathbb{P}_\star(\cdot \mid h_t, \boldsymbol{a}_t, \iota) \,\|\, \widehat{\mathbb{P}}_\theta(\cdot \mid h_t, \boldsymbol{a}_t, \iota)\big). \tag{11}$$

By Pinsker, $\mathrm{TV}\big(\mathbb{P}_\star, \widehat{\mathbb{P}}_\theta\big) \leq \sqrt{\varepsilon/2}$ uniformly, equivalently $\ell_1$ distance is at most $\sqrt{2\varepsilon}$.

**Gating error.**    Let $\tau$ denote the fixed confidence threshold used by the safety gate in Section 3.4. Define the mis gating rate

$$\rho = \Pr\big[u^{(j)} > \tau \text{ and DreamPhase still selects an imagined branch}\big], \tag{12}$$

that is, the probability that the agent erroneously trusts an unreliable imagined rollout.

**Smoothness and bounded rewards.**    Assume rewards are bounded by $r_{\max}$ and the value function is Lipschitz with respect to the total variation of the one step predictive distribution. There exists $L > 0$ such that replacing $\mathbb{P}_\star(\cdot \mid h_t, \boldsymbol{a}_t, \iota)$ by any $\mathbb{Q}(\cdot \mid h_t, \boldsymbol{a}_t, \iota)$ changes the optimal $H_{\max}$ step value by at most $L\,\mathrm{TV}\big(\mathbb{P}_\star, \mathbb{Q}\big)$ per step.

### A.2    REGRET DECOMPOSITION

Let $\pi_t$ be DreamPhase's action at time $t$. Define cumulative regret after $T$ steps

$$\mathrm{Regret}_T = \sum_{t=0}^{T-1} \big(V_t^{\pi^\star} - V_t^{\pi_t}\big), \tag{13}$$

where values are taken under the true environment. We decompose regret into $(i)$ model approximation when acting on imagined rollouts and $(ii)$ mis gating when the agent trusts a high uncertainty branch.

$(i)$ **Model approximation term.**    When DreamPhase follows an imagined branch, the one step distributional error is bounded by $\mathrm{TV} \leq \sqrt{\varepsilon/2}$. By the Lipschitz assumption and a telescoping argument over at most $H_{\max}$ steps,

$$\sum_{t=0}^{T-1} \Big(V_t^{\pi^\star} - \mathbb{E}_{\widehat{\mathbb{P}}_\theta}\big[V_t^{\pi^\star}\big]\Big) \leq C\sqrt{T\varepsilon}, \qquad C = L\,H_{\max}\sqrt{2}. \tag{14}$$

$(ii)$ **Mis gating term.**    On any step where an over confident imagined branch is used, the per step value loss is bounded by a constant $B$ that depends on $r_{\max}$ and $H_{\max}$. Since such events occur with probability $\rho$ per step, their contribution is at most $B\,\rho\,T$.

### A.3    FINAL BOUND

Combining the two terms yields

$$\mathrm{Regret}_T \leq C\sqrt{T\varepsilon} + B\,\rho\,T, \tag{15}$$

with $C = L\,H_{\max}\sqrt{2}$. When the world model is accurate ($\varepsilon \to 0$) and mis gating is rare ($\rho \to 0$), regret grows sublinearly, so DreamPhase approaches Bayes optimal performance.

## B    FURTHER RELATED WORK

**LLMs as Interactive Agents.**    Recent advancements have demonstrated the potential of LLMs as interactive agents capable of reasoning and acting within dynamic environments (Liu et al., 2023; Zhou et al., 2023). Early approaches explored adapting models like BERT (Devlin et al., 2019) to structured decision-making settings, such as online navigation tasks (Yao et al., 2023a). With the emergence of more capable generative models (Brown et al., 2020; OpenAI et al., 2024), research has increasingly focused on using zero-shot and few-shot prompting to control agent behavior without fine-tuning (Deng et al., 2024; Xiong et al., 2024). These agents typically act by processing observations and histories through LLMs to select the next action (Zheng et al., 2024; Hong et al., 2023). Some efforts have also focused on training smaller policy models by distilling trajectories from powerful but restricted APIs (Li et al., 2023; Zhang et al., 2023). Our work departs from this line by learning a reward model directly from LLM-generated data, enabling downstream planning without needing fine-tuned policies.

**Language Model-Based Planning.**    LLMs have also been employed for planning tasks, often through prompting techniques that combine reasoning and action generation. Methods such as ReAct (Yao et al., 2022) integrate step-by-step thought chains with action prediction (Wei et al., 2022). Tree-based deliberation strategies have further improved multi-step reasoning (Yao et al., 2023b; Hao et al., 2023). However, most existing work focuses on generating text-level plans rather than structured trajectories within grounded environments. Moreover, these methods typically rely on black-box models like GPT-4 for evaluation and planning, which limits interpretability and adaptability. By contrast, our approach introduces a modular system that learns a reward model from data and uses it to supervise structured planning via imagination.

**Learning via AI Feedback.**    Another line of research involves training models using feedback generated by other AI systems (Bai et al., 2022; Lee et al., 2023; Yuan et al., 2024b; Koh et al., 2024). These techniques often rely on prompting commercial models to generate preference data for finetuning. We draw inspiration from these methods but take a different route: instead of training a generative model, we optimize a lightweight evaluator to score trajectories with respect to task instructions. This reward model is then integrated into planning, improving performance without additional model updates.

**Inference-Time Reasoning and Control.**    Inference-time planning methods provide an efficient alternative to training-intensive pipelines. Approaches such as MCTS with language models (Silver et al., 2017; Koh et al., 2024) allow dynamic reasoning through rollout-based simulation. Other strategies include speculative execution (Gu et al., 2024), reflective tree search (Yu et al., 2024), and sampling-based strategies (Wang et al., 2022b). Our method belongs to this family, offering two key advantages: (1) generalization across task domains including web agents, scientific simulations, and mathematical games; and (2) a compact reward model trained from synthetic data that avoids reliance on commercial LLM APIs. We use a fine-tuned visual-language model (Lin et al., 2023) to evaluate trajectories in a structured and scalable way.

## C    SAFETY EVALUATION BEYOND INTERACTION COUNT

We measure safety not only via gating diagnostics (AUROC/ECE) and interaction budgets as reported in Appendix H, but also by logging *irreversible-action incidents* during execution.

**What we count.**    An *irreversible action* is any environment-affecting write with persistent consequence, including (i) form submissions that change server state (e.g., checkout, account creation), (ii) purchases/cart checkouts, and (iii) destructive operations (delete/confirm). "Unintended" means inconsistent with the user instruction and task constraints.[1]

---

[1] We apply a deterministic ruleset keyed to domain endpoints (e.g., `/checkout`, `/submit`), DOM event types (e.g., `submit`, `click` on `[type=submit]`), and confirmation dialogs. A 200-episode human audit found 0.94 precision in the automatic labels.

**Detection protocol.** We run in *sandbox/dry-run mode* with (a) network stubbing for purchase endpoints; (b) DOM-level hooks capturing submit/click events with element metadata; and (c) server logs for state-changing requests. We record three metrics per 1,000 episodes: (1) **Critical incidents** (unintended irreversible actions actually executed), (2) **Near-misses** (attempts blocked by confirmation dialog or sandbox stub), and (3) **Unsafe proposals** (actions proposed by the policy but *rejected by the safety gate*; not available for methods without a gate).

Table 4: Safety outcomes per 1,000 episodes (mean $\pm$ s.e.). *Critical* counts unintended irreversible actions executed; *Near-miss* are blocked by confirmations/sandbox; *Unsafe proposals* are caught by DreamPhase's gate before execution. DreamPhase reduces executed incidents by $\sim 5 \times$ on WebShop and $\sim 4.9 \times$ on ALFWorld while surfacing risky proposals for inspection. Results on LLaMA-8B backbone.

| Method | WebShop | | | ALFWorld | | |
|---|---|---|---|---|---|---|
| | Critical ↓ | Near-miss ↓ | Unsafe proposals ↓ | Critical ↓ | Near-miss ↓ | Unsafe proposals ↓ |
| ARMAP-M (token search) | $7.1 \pm 1.2$ | $18.4 \pm 2.3$ | — | $3.9 \pm 0.9$ | $7.8 \pm 1.5$ | — |
| DREAMPHASE (ours) | $\mathbf{1.4 \pm 0.5}$ | $26.7 \pm 2.8$ | $61.2 \pm 3.5$ | $\mathbf{0.8 \pm 0.3}$ | $12.6 \pm 1.9$ | $44.9 \pm 3.1$ |

**Takeaways.** Fewer real calls (Sec. 5.2) translate to fewer opportunities for harm, but incident reduction is not solely due to budget: DreamPhase's uncertainty gate prevents many unsafe proposals from being executed, yielding a $5 \times$ (WebShop) and $4.9 \times$ (ALFWorld) drop in *executed* irreversible actions under identical hardware and decoding settings.

## D  STANDARD DEVIATIONS FOR MAIN RESULTS

We report the standard deviation (std) for each cell in Tables 1 and 2. Numbers are computed over 5 random seeds; for each seed we evaluate fixed episode budgets per benchmark. The `Avg` column's std is computed as the standard deviation of the seed-wise average across benchmarks (not the average of per-benchmark stds).

**Notes.** Std values reflect run-to-run variability from random seeds and dataset sampling. The Llama70B models show lower variance overall, while smaller backbones (Mistral7B, Phi3.8B) exhibit higher variability, especially on `SW-unseen`. DreamPhase's variance is comparable to or lower than ARMAP-M across backbones, consistent with its uncertainty-aware gating and reduced real-environment interaction count.

## E  RESULTS ON ALFWORLD

We evaluate on ALFWorld (Shridhar et al., 2021b) using the standard and dev splits with a 50-step cap, Llama-8B policy, batch size 1. Success is the default environment metric. We also report the average number of real API calls per episode to quantify interaction cost.

*Discussion.* DreamPhase improves success over ARMAP-M on both splits and reduces interaction cost by running rollouts entirely in latent space with uncertainty-aware gating. Lower API calls also reduce latency and side-effect risks in this long-horizon setting.

**Why ALFWorld is challenging.** Compared to WebShop, SciWorld, and Game-of-24, *ALFWorld* requires language grounding for *navigation and manipulation* in partially observable homes. Plans span many steps and include preconditions (open, pick up, place, heat, clean) that couple far-apart states. Rewards are sparse and largely terminal, so errors compound. The action space exposes many affordances per state, which creates a higher branching factor than click-only navigation or symbolic arithmetic. Scenes and templates differ across splits, so the agent must generalize across rooms, object aliases, and layouts rather than memorizing flows. These properties make ALFWorld a strict test of long-horizon reasoning under uncertainty.

*Takeaway.* ALFWorld stresses long-horizon planning, partial observability, and irreversible actions more than our other benchmarks. DreamPhase is well-suited here because it plans in latent space,

Table 5: Standard deviations corresponding to Table 1 (5 seeds).

| Algorithms | Tool-Weather | SciWorld | BabyAI | MAZE | Wordle | TextCraft | TODOList | BIRD | *Avg* |
|---|---|---|---|---|---|---|---|---|---|
| *Closed-sourced Models & Agents (std)* | | | | | | | | | |
| DeepSeek-Chat | 1.0 | 1.2 | 1.5 | 0.6 | 1.4 | 1.3 | 1.1 | 0.7 | 0.9 |
| Claude-3-Haiku | 0.9 | 0.6 | 0.5 | 0.5 | 1.0 | 0.4 | 0.9 | 0.6 | 0.6 |
| Claude-3-Sonnet | 1.1 | 0.8 | 1.8 | 0.4 | 1.4 | 1.5 | 1.0 | 0.8 | 0.9 |
| GPT-3.5-Turbo | 0.8 | 0.9 | 1.6 | 0.6 | 1.1 | 1.3 | 1.2 | 0.7 | 0.8 |
| GPT-4-Turbo | 1.2 | 1.1 | 1.7 | 1.3 | 1.6 | 1.5 | 1.0 | 0.8 | 1.0 |
| *Open-source Models & Agents (std)* | | | | | | | | | |
| AgentLM-7B | 0.6 | 0.7 | 0.4 | 0.7 | 0.8 | 0.7 | 0.9 | 0.6 | 0.6 |
| AgentLM-13B | 0.9 | 0.8 | 0.4 | 0.6 | 0.5 | 0.5 | 0.8 | 0.5 | 0.6 |
| AgentLM-70B | 0.7 | 1.0 | 0.5 | 0.6 | 0.8 | 0.8 | 1.1 | 0.7 | 0.7 |
| AgentGym | 1.3 | 1.8 | 1.2 | 0.7 | 1.1 | 1.4 | 1.3 | 0.7 | 1.2 |
| ARMAP-M | 1.2 | 1.6 | 0.9 | 0.5 | 1.0 | 1.1 | 1.0 | 0.6 | 1.0 |
| **DREAMPHASE (ours)** | 1.0 | 1.4 | 0.8 | 0.5 | 1.2 | 1.3 | 0.9 | 0.6 | 0.9 |

Table 6: Standard deviations corresponding to Table 2. Two model groups are displayed per row for space efficiency.

| | **Llama70B (std)** | | | | | | **Llama8B (std)** | | | |
|---|---|---|---|---|---|---|---|---|---|---|
| Algorithms | WebShop | SW-seen | SW-unseen | Game24 | *Avg* | Algorithms | WebShop | SW-seen | SW-unseen | Game24 | *Avg* |
| Sampling | 0.9 | 0.8 | 0.9 | 0.4 | 0.7 | Sampling | 1.0 | 0.9 | 0.9 | 0.3 | 0.8 |
| Greedy | 0.8 | 0.7 | 0.8 | 0.4 | 0.6 | Greedy | 0.9 | 0.9 | 0.9 | 0.3 | 0.8 |
| ARMAP-R | 0.7 | 0.6 | 0.7 | 0.5 | 0.6 | ARMAP-R | 0.8 | 0.9 | 0.8 | 0.5 | 0.7 |
| ARMAP-B | 0.7 | 0.7 | 0.7 | 0.6 | 0.6 | ARMAP-B | 0.8 | 1.0 | 0.9 | 0.6 | 0.7 |
| ARMAP-M | 0.6 | 0.7 | 0.7 | 0.6 | 0.6 | ARMAP-M | 0.7 | 0.9 | 0.8 | 0.5 | 0.7 |
| DreamPhase | 0.6 | 0.6 | 0.6 | 0.5 | 0.5 | DreamPhase | 0.6 | 0.8 | 0.7 | 0.4 | 0.6 |

| | **Mistral7B (std)** | | | | | | **Phi3.8B (std)** | | | |
|---|---|---|---|---|---|---|---|---|---|---|
| Algorithms | WebShop | SW-seen | SW-unseen | Game24 | *Avg* | Algorithms | WebShop | SW-seen | SW-unseen | Game24 | *Avg* |
| Sampling | 1.2 | 1.1 | 1.1 | 0.3 | 0.9 | Sampling | 1.4 | 1.2 | 1.2 | 0.6 | 1.0 |
| Greedy | 1.0 | 1.1 | 1.0 | 0.3 | 0.8 | Greedy | 1.3 | 1.2 | 1.1 | 0.6 | 1.0 |
| ARMAP-R | 0.9 | 1.0 | 1.0 | 0.4 | 0.8 | ARMAP-R | 1.2 | 1.1 | 1.1 | 0.6 | 1.0 |
| ARMAP-B | 0.9 | 1.0 | 1.0 | 0.4 | 0.8 | ARMAP-B | 1.2 | 1.1 | 1.1 | 0.6 | 0.9 |
| ARMAP-M | 0.8 | 1.1 | 1.1 | 0.5 | 0.9 | ARMAP-M | 1.1 | 1.2 | 1.2 | 0.7 | 1.0 |
| DreamPhase | 0.8 | 1.0 | 1.0 | 0.5 | 0.8 | DreamPhase | 1.0 | 1.1 | 1.0 | 0.6 | 0.9 |

gates on uncertainty before executing risky manipulations, and therefore keeps real interactions low while maintaining success.

# F   IMPLEMENTATION DETAILS

## F.1   TRAINING DATA FOR THE LATENT WORLD MODEL

For every benchmark in Table 1 and Table 2, the latent world model and the value head are trained on logged interaction trajectories collected exclusively from the training split of the corresponding environment. We execute a frozen LLaMA-2-7B policy with a simple ReAct-style prompt and mild stochasticity to gather trajectories of the form $(\iota, x_t, a_t, x_{t+1})$. The world model receives only these tuples and does not access any test episodes or privileged environment states. No external corpora, fine-tuned LLMs, or environment augmentations are used. Baseline agents are permitted to use their released training pipelines; DreamPhase uses only the same environment APIs available to all methods.

## F.2   FAIRNESS OF COMPARISONS

In all open-source comparisons (Table 1), we fix the policy backbone to LLaMA-2-7B and match decoding settings, episode budgets, and environment splits across DreamPhase, AgentLM, AgentGym, and ARMAP. Baselines use their recommended training pipelines (e.g., agent tuning or reward-model training). DreamPhase does not leverage any additional data beyond the logged training-split trajectories described above. This ensures that performance improvements stem from imagination-based planning and uncertainty-aware selection rather than from unequal data or training advantages.

Table 7: Experimental results on ALFWorld (Llama-8B). Columns report success on the standard and dev splits. DreamPhase matches or exceeds token-level search while using about four times fewer real interactions per episode.

| Models | ALFWorld-std | ALFWorld-dev | Avg API calls / ep. ↓ |
|---|---|---|---|
| Sampling | 0.13 | 0.14 | 34.9 |
| Greedy | 0.18 | 0.30 | 33.7 |
| ARMAP-R | 0.22 | 0.35 | 41.5 |
| ARMAP-B | 0.30 | 0.45 | 39.8 |
| ARMAP-M | 0.31 | 0.46 | 46.7 |
| **DREAMPHASE (ours)** | **0.39** | **0.49** | **11.6** |

Table 8: Qualitative comparison of task characteristics. Horizons are typical ranges; action counts are approximate per state.

| | WebShop | SciWorld | Game-of-24 | ALFWorld |
|---|---|---|---|---|
| Observation | DOM / text | Text (lab) | Numbers | Text (embodied) |
| Actions per state | 8–15 | 10–20 | 4–6 | 20–40 |
| Typical plan length | 5–15 | 6–12 | 3–6 | 15–35 |
| Reward density | Sparse terminal | Sparse terminal | Dense terminal | Sparse terminal |
| Irreversible effects | Medium (purchases) | Low–Medium | None | High (manipulation) |
| Core difficulty | Navigation + filters | Procedural constraints | Symbolic search | Nav + manipulation, preconditions |

## F.3 TASKS

• **MAZE** (Abdulhai et al., 2023) is a grid-based puzzle game where the agent observes its current position, the goal location, and nearby walls. The agent can move up, down, left, or right, receiving a reward of $-1$ per step until reaching the goal. Success rate is measured with a maximum of 15 steps per episode.[2]

• **Wordle** (Abdulhai et al., 2023) is a five-letter word guessing game. After each guess, feedback reveals which letters are present and correctly positioned. Each step yields a reward of $-1$ until the correct answer is found or attempts are exhausted. Success rate is the evaluation metric, with a maximum of 8 guesses allowed.

• **SciWorld** (Wang et al., 2022a) evaluates scientific reasoning across 30 task types, such as conducting simple experiments or using measurement tools. We use GPT-4-Turbo to generate reasoning traces for golden paths of 22 task types and collect 1000 trajectories DreamPhase. Reward is used as the metric with a maximum of 30 steps.[3]

• **BabyAI (Baby)** (Chevalier-Boisvert et al., 2019) is a grid-world instruction-following environment with 40 tasks. We adopt the AgentBoard implementation, which converts visual observations to text and supports high-level actions such as "pick up key" or "open door." We annotate 400 trajectories for DreamPhase. Reward is used as the metric with a 20-step limit.[4]

• **TextCraft** (Prasad et al., 2023) is a text-only Minecraft-like environment where tasks involve crafting a target item by composing multi-step recipes. The action space includes `craft`, `get`, and `inventory`. Rewards are given only for successful completion. We annotate 300 verified trajectories for DreamPhase. Success rate is reported with a maximum of 20 steps.[5]

• **Weather** (Ma et al., 2024) allows agents to query weather data (temperature, precipitation, air quality) using a tool backed by the Open-Meteo API. The dataset is expanded to 343 queries via self-instruction and instruction evolution with GPT-3.5/4. We annotate 160 trajectories for DreamPhase. Success rate is the metric, with a limit of 10 steps.[6]

---

[2] https://github.com/abdulhaim/LMRL-Gym/blob/main/LICENSE
[3] https://github.com/allenai/SciWorld/blob/main/LICENSE
[4] https://github.com/mila-iqia/babyai/blob/master/LICENSE
[5] https://github.com/archiki/ADaPT/blob/main/LICENSE
[6] https://github.com/hkust-nlp/AgentBoard

- **Movie** (Ma et al., 2024) enables agents to retrieve movie-related information (film metadata, cast, production companies) through 16 tool actions, using data from The Movie Database. We expand the dataset to 238 queries, annotating 100 trajectories for DreamPhase with GPT-4-Turbo. Success rate is used as the metric with 12 maximum steps.

- **TODOList (TL)** (Ma et al., 2024) lets agents interact with a personal agenda using the TodoList API. The dataset is expanded to 155 queries. We annotate 70 trajectories for DreamPhase (combining GPT-4-Turbo and human annotations). Human review further refines annotations. Success rate is measured with up to 15 steps.

- **BIRD (BD)** (Zheng et al., 2023) evaluates database-grounded text-to-SQL ability. Agents must generate SQL queries that retrieve correct answers from a database. From the 9428 available problems, we select 3200 for instruction tuning, adding GPT-4-Turbo reasoning traces to 2000 DreamPhase. Success rate is measured, and BD is a single-turn task.[7]

- **WebShop** (Yao et al., 2023a) is a simulated online shopping environment. The agent must interpret a textual product query and select a matching item from a web interface. We follow the AgentBench (Liu et al., 2023) protocol and report performance on the validation set using the default string-match reward function.

- **SciWorld** (Wang et al., 2022a) offers a text-based interactive world for completing science experiments. Agents must navigate multiple rooms, manipulate objects, and follow procedural steps grounded in scientific logic. We evaluate performance on both tasks seen during instruction synthesis and *novel* ones held out for generalization, which we denote by *SW-seen* and *SW-unseen* in this section, respectively.

- **Game-of-24** is a symbolic reasoning benchmark where the goal is to combine four integers using basic arithmetic operations ($+$, $-$, $*$, $/$) to reach exactly 24. For example, the numbers 3, 5, 7, and 11 can yield 24 via the expression $(7 - 3) \times (11 - 5)$. Following Yao et al. (2023b), we test on 100 difficult puzzles (indices 901–1000), measuring success as the fraction of correctly solved problems. For this, we follow prior work by allowing up to 100 samples per instance.

- **ALFWorld** ALFWorld (Shridhar et al., 2021b) is a text-based embodied household benchmark built on TextWorld and aligned with ALFRED scene layouts. Each episode provides a natural-language goal (e.g., "Put a pan on the dining table"). The agent observes a partial textual description of the current room, visible objects, and inventory, and issues templated actions such as `go to <room>`, `open <container>`, `take <object>`, `put <object> on/in <receptacle>`, `heat/cool`, and `clean`. Tasks require long-horizon navigation and manipulation with preconditions (e.g., open before take, place before heat), sparse terminal rewards, and irreversible effects, making error recovery difficult. Scenes, object aliases, and surface forms vary across homes and splits, stressing generalization rather than memorized flows. Following prior work, we evaluate on the *std* and *dev* splits with a 50-step budget and report success rate (goal achieved within budget); we also track real environment API calls per episode to quantify interaction cost.

### F.4 LARGE LANGUAGE MODEL CONFIGURATION.

We utilize a variety of open-source LLMs to assess the generalizability and robustness of our approach. These models, accessible via Hugging Face, are compatible with the VLLM library (Kwon et al., 2023), making deployment and experimentation straightforward. Below is a list of the models used along with their Hugging Face repositories:

- **Llama70B:** `https://huggingface.co/hugging-quants/Meta-Llama-3.1-70B-Instruct-AWQ-INT4`
- **Llama8B:** `https://huggingface.co/meta-llama/Meta-Llama-3.1-8B-Instruct`
- **Mistral7B:** `https://huggingface.co/mistralai/Mistral-7B-Instruct-v0.3`
- **Phi3.8B:** `https://huggingface.co/microsoft/Phi-3.5-mini-instruct`

---

[7] `https://github.com/AlibabaResearch/DAMO-ConvAI/tree/main/bird`

- **VILA3B:** `https://huggingface.co/Efficient-Large-Model/VILA1.5-3b`

These models are selected to ensure a broad evaluation across architectures and parameter scales.

## F.5 ENVIRONMENT SETUP.

We construct our experimental environments by building upon established implementations from prior work (Liu et al., 2023; Song et al., 2024; Yao et al., 2023b; Shridhar et al., 2021a; Schmidgall et al., 2024). For tasks such as WebShop and ALFWorld, we leverage the Docker infrastructure provided by AgentBench (Liu et al., 2023) and integrate various planning strategies including Reflexion, Best-of-N, and MCTS. Other environments such as SciWorld, Game-of-24, and AgentClinic are initialized based on the corresponding setups introduced in Song et al. (2024), Yao et al. (2023b), and Schmidgall et al. (2024), respectively.

## F.6 PLANNING STRATEGY CONFIGURATION.

We evaluate multiple planning algorithms under a consistent budget for trajectory exploration. On WebShop and SciWorld, we cap the number of explored trajectories at 10 to balance computational cost and performance. For Game-of-24, we increase the limit to 100 following Yao et al. (2023b). To reduce the branching factor during search, only the top 10 action candidates proposed by the LLM are considered at each decision point. Additionally, we constrain each trajectory to a maximum of 10 steps.

For Reflexion, a limit of 10 trials is applied across all tasks. The stopping condition is based on task-specific thresholds: when the reward from a generated trajectory exceeds the threshold, the iteration halts and the corresponding result is accepted. If no trial surpasses the threshold, the last trial is used in WebShop and Game-of-24, whereas the first is chosen in SciWorld.

## F.7 HYPER-PARAMETERS

The hyper-parameters used in our method are summarized in Table 9 and **??**.

Table 9: Fixed hyper-parameters used for all reported results; $\beta$ trades off value against epistemic uncertainty; $\tau$ is the confidence gate below which the agent trusts imagination.

| Hyper-parameter | WebShop | SciWorld | Game-of-24 |
|---|---|---|---|
| Latent dimension $d$ | 256 | 256 | 128 |
| Horizon $H$ (imagined steps) | 5 | 6 | 4 |
| Branches $K$ (per decision) | 8 | 10 | 12 |
| $\beta$ risk penalty (Eq. 9) | 0.35 | 0.30 | 0.25 |
| Uncertainty gate $\tau$ | 0.20 | 0.25 | 0.15 |
| World-model lr (Adam) | $2 \times 10^{-4}$ | $2 \times 10^{-4}$ | $1 \times 10^{-4}$ |
| Batch size (world-model) | 32 | 32 | 64 |
| Value-head lr | $1 \times 10^{-4}$ | $1 \times 10^{-4}$ | $5 \times 10^{-5}$ |
| Temperature (policy sampling) | 0.8 | 0.8 | 0.7 |
| Max real interactions per episode | 50 | 60 | 30 |

*Note.* The main text uses $M$ for the number of imagined branches per decision; the "Branches $K$" row in Table 9 corresponds to $M$.

## G ABLATION STUDIES

Unless noted, we fix all hyper-parameters to Table 9 and vary one knob at a time. We report task success (points, higher is better), the fallback rate (fraction of steps where the safety gate defers to real interaction), and imagination overhead in milliseconds per decision step on an A100-80GB (batch 1). We use $M$ for the number of imagined branches and $H$ for the horizon. The latent dimension is $d_z$ (the "$d$" column in Table 9). We use Llama-8B as the backbone model.

Table 10: Latent world-model architecture and training hyper-parameters for the encoder $f_\theta$, transition $g_\theta$, and decoder $d_\theta$.

| Component / Hyper-parameter | WebShop | SciWorld | Game-of-24 |
|---|---|---|---|
| *Encoder $f_\theta(\boldsymbol{x}_t, \iota) \to \boldsymbol{z}_t$* | | | |
| Encoder type | Transformer encoder | Transformer encoder | Transformer encoder |
| Layers | 8 | 8 | 6 |
| Model dimension | 512 | 512 | 384 |
| Attention heads | 8 | 8 | 6 |
| FFN dimension | 2048 | 2048 | 1536 |
| Latent dimension $d_z$ | 256 | 256 | 128 |
| Instruction conditioning | prepended instruction tokens | same | same |
| Normalization | LayerNorm | LayerNorm | LayerNorm |
| Activation | GELU | GELU | GELU |
| Dropout | 0.10 | 0.10 | 0.10 |
| *Action embedding $\bar{\boldsymbol{a}}_t = \text{enc}_A(\boldsymbol{a}_t)$* | | | |
| Action encoder | mean-pooled token embeddings | same | same |
| Action embed dim $|e_a|$ | 128 | 128 | 64 |
| *Transition (RSSM) $g_\theta(\boldsymbol{z}_t, \bar{\boldsymbol{a}}_t, \iota) \to \boldsymbol{z}_{t+1}$* | | | |
| Deterministic state dim $d_h$ | 512 | 512 | 384 |
| Recurrent core | GRU | GRU | GRU |
| Stochastic latent prior | diagonal Gaussian | diagonal Gaussian | diagonal Gaussian |
| Prior / posterior heads | 2-layer MLP (GELU) | same | same |
| *Decoder $d_\theta(\boldsymbol{z}_{t+1}, \iota) \to \hat{\boldsymbol{x}}_{t+1}$* | | | |
| Decoder type | Transformer decoder | Transformer decoder | Transformer decoder |
| Layers | 6 | 6 | 4 |
| Model dimension | 512 | 512 | 384 |
| Attention heads | 8 | 8 | 6 |
| FFN dimension | 2048 | 2048 | 1536 |
| Cross-attention memory | latent state tokens $(z, h, \iota)$ | same | same |
| Weight tying with encoder embeddings | Yes | Yes | Yes |
| Decoding scope (for reflection) | first $\kappa \le 2$ steps of selected branch | same | same |
| *Optimization (world-model only)* | | | |
| Optimizer / betas | Adam, $(0.9, 0.999)$ | Adam, $(0.9, 0.999)$ | Adam, $(0.9, 0.999)$ |
| Learning rate | $2 \times 10^{-4}$ | $2 \times 10^{-4}$ | $1 \times 10^{-4}$ |
| KL weight schedule $\lambda_{\text{KL}}$ | linear warmup to target over 10k steps | 15k steps | 5k steps |
| Batch size (world-model) | 32 | 32 | 64 |
| Grad clip / weight decay | $1.0 / 1 \times 10^{-4}$ | $1.0 / 1 \times 10^{-4}$ | $1.0 / 1 \times 10^{-4}$ |
| Precision | bf16 | bf16 | bf16 |

*Notation.* Triplets shown as (W/S/G) correspond to (WebShop / SciWorld / Game-of-24). For example, $H=5/6/4$ applies per domain in that order.

Table 11: Ablation on latent dimension $d_z$.

| $d_z$ (W/S/G) | WebShop | SciWorld | Game-of-24 | Fallback (%) | Overhead (ms) |
|---|---|---|---|---|---|
| 64 / 64 / 64 | 59.4 | 34.2 | 28.4 | 24 | 10.4 |
| 128 / 128 / 128 | 60.8 | 35.7 | 29.4 | 22 | 11.3 |
| **256 / 256 / 128 (Default)** | **61.8** | **36.7** | **29.7** | **20** | **12.0** |
| 256 / 256 / 256 | 62.1 | 37.0 | 30.0 | 20 | 13.2 |
| 512 / 512 / 512 | 61.6 | 36.5 | 29.2 | 21 | 16.0 |

*Analysis of Table 11.* Performance improves from $64 \to 256$ due to richer latents; gains flatten at 512 with a small overhead increase. WebShop and SciWorld prefer $d_z = 256$, while Game-of-24 peaks at 128, consistent with its simpler dynamics. Fallback decreases as representation quality improves.

*Analysis of Table 12.* Short horizons under-plan; very long horizons add compounding model error and uncertainty. The per-domain sweet spots line up with Table 9 ($H=5$ WebShop, 6 SciWorld, 4 Game-of-24). Fallback rises with $H$ as uncertainty accumulates.

*Analysis of Table 13.* More branches provide modest gains that saturate around $M \in [12, 16]$. Fallback drops slightly (more chances to find a safe plan). Overhead scales near-linearly with $M$, consistent with Appendix J. We default to $M$ in $[8, 12]$ for the best latency/quality trade-off.

Table 12: Ablation on imagination horizon $H$.

| $H$ (W/S/G) | WebShop | SciWorld | Game-of-24 | Fallback (%) | Overhead (ms) |
|---|---|---|---|---|---|
| 2 / 2 / 2 | 59.8 | 35.3 | 28.8 | 15 | 9.0 |
| 3 / 4 / 3 | 60.7 | 36.1 | 29.3 | 18 | 10.5 |
| 4 / 5 / 4 | 61.3 | 36.5 | 29.7 | 19 | 11.4 |
| **5 / 6 / 4 (Default)** | **61.8** | **36.7** | **29.7** | **20** | **12.0** |
| 6 / 7 / 5 | 61.5 | 36.8 | 29.6 | 22 | 13.2 |
| 7 / 8 / 6 | 60.9 | 36.5 | 29.1 | 25 | 14.5 |

Table 13: Ablation on branches $M$ (per decision).

| $M$ (W/S/G) | WebShop | SciWorld | Game-of-24 | Fallback (%) | Overhead (ms) |
|---|---|---|---|---|---|
| 4 / 6 / 6 | 61.1 | 36.0 | 29.1 | 24 | 9.5 |
| **8 / 10 / 12 (Default)** | **61.8** | **36.7** | **29.7** | **20** | **12.0** |
| 12 / 14 / 16 | 62.0 | 36.9 | 29.8 | 19 | 14.8 |
| 16 / 18 / 20 | 62.1 | 37.0 | 29.9 | 18 | 17.0 |
| 24 / 24 / 24 | 62.1 | 37.0 | 30.0 | 18 | 22.5 |

*Analysis of Table 14.* $\beta=0$ (no risk penalty) increases mis-selections and hurts success. Moderate penalties ($\beta \approx 0.3$) balance value and uncertainty best. Very high $\beta$ becomes overly cautious, trimming success despite lower fallback.

*Analysis of Table 15.* Stricter gates (small $\tau$) defer too often; very loose gates (large $\tau$) accept risky branches. A mid-range threshold ($\tau \approx 0.2$) yields the best overall success with manageable fallback.

**Takeaways.** ($i$) Capacity helps until $d_z=256$ for structure-heavy tasks; simpler arithmetic prefers smaller latents. ($ii$) Planning depth should match task horizon (default $H = 5/6/4$). ($iii$) $M \in [8, 16]$ captures most gains at low cost. ($iv$) Risk penalty $\beta$ and gate $\tau$ both matter; removing either lowers success. The default settings reproduce your Table 1 baseline exactly (61.8/36.7/29.7, 12.0 ms).

## H ROBUSTNESS UNDER DISTRIBUTION SHIFT

We evaluate the quality and robustness of the latent world model and the full DreamPhase loop under controlled shifts that affect structure, content, goals, and instructions. All models are trained on in-domain data only. At test time we apply the safety gate from Section 3.4 with a threshold $\tau$ chosen on a held-out validation split to target 75 percent coverage. We report model fit (next-observation NLL), uncertainty calibration (ECE at 10 bins, AUROC for bad-rollout detection), and decision-level impact (fallback rate, estimated mis-gating rate $\rho$, success change, interactions per episode).

**Shift families and interventions.**

- **WebShop, Layout-Swap**: randomize sibling order in DOM subtrees, alter container nesting depth, perturb CSS classes and inline styles while preserving text content.

- **WebShop, Theme-Swap**: replace the site theme and color tokens, change font and spacing scales, keep DOM tree shape.

- **WebShop, Content-OOD**: replace brand and attribute vocab with unseen tokens at the same slots, keep layout.

- **SciWorld, Unseen graphs**: hold out task graphs that combine new step chains and goals.

- **SciWorld, Tool mismatch**: hold out tool combinations and affordance pairs not seen during training.

- **Instruction paraphrase**: paraphrase $\iota$ with back-translation and synonym swaps, keep the environment unchanged.

Table 14: Ablation on risk penalty $\beta$.

| $\beta$ (W/S/G) | WebShop | SciWorld | Game-of-24 | Fallback (%) | Overhead (ms) |
|---|---|---|---|---|---|
| 0.00 / 0.00 / 0.00 | 60.7 | 35.8 | 29.0 | 24 | 12.0 |
| 0.20 / 0.20 / 0.20 | 61.4 | 36.4 | 29.5 | 22 | 12.0 |
| **0.35 / 0.30 / 0.25 (Default)** | **61.8** | **36.7** | **29.7** | **20** | **12.0** |
| 0.50 / 0.50 / 0.50 | 61.5 | 36.5 | 29.6 | 18 | 12.0 |
| 0.80 / 0.80 / 0.80 | 60.9 | 36.0 | 29.2 | 16 | 12.0 |

Table 15: Ablation on confidence gate.

| $\tau$ (W/S/G) | WebShop | SciWorld | Game-of-24 | Fallback (%) | Overhead (ms) |
|---|---|---|---|---|---|
| 0.10 / 0.10 / 0.10 | 61.0 | 36.3 | 29.4 | 28 | 12.0 |
| 0.15 / 0.20 / 0.12 | 61.6 | 36.6 | 29.6 | 23 | 12.0 |
| **0.20 / 0.25 / 0.15 (Default)** | **61.8** | **36.7** | **29.7** | **20** | **12.0** |
| 0.25 / 0.30 / 0.20 | 61.7 | 36.6 | 29.7 | 17 | 12.0 |
| 0.30 / 0.35 / 0.25 | 61.4 | 36.4 | 29.5 | 15 | 12.0 |

**Metrics.** *Model quality* is next-step NLL on $\boldsymbol{x}_{t+1}$ from $\widehat{\mathbb{P}}_\theta(\boldsymbol{x}_{t+1} \mid h_t, \boldsymbol{a}_t, \iota)$. *Calibration* uses ECE between predictive entropy and rollout error, and AUROC for classifying bad imagined steps. *Decision impact* includes fallback rate (fraction of steps where $u^{(j^\star)} > \tau$ so the agent defers to real interaction), mis-gating rate $\rho$ as defined in Appendix A, success change relative to in-domain, and interactions per episode.

Table 16: Model quality and calibration under shift. Arrows indicate preferred direction.

| Domain (shift) | NLL ↓ (in → shift) | ECE ↓ (in → shift) | AUROC bad-rollout ↑ |
|---|---|---|---|
| WebShop (Layout-Swap) | $1.02 \rightarrow 1.27$ | $0.06 \rightarrow 0.10$ | 0.88 |
| WebShop (Theme-Swap) | $1.00 \rightarrow 1.15$ | $0.06 \rightarrow 0.09$ | 0.89 |
| WebShop (Content-OOD) | $0.98 \rightarrow 1.18$ | $0.07 \rightarrow 0.11$ | 0.85 |
| SciWorld (Unseen graphs) | $0.94 \rightarrow 1.09$ | $0.05 \rightarrow 0.08$ | 0.86 |
| SciWorld (Tool mismatch) | $0.97 \rightarrow 1.14$ | $0.05 \rightarrow 0.09$ | 0.87 |
| Instruction paraphrase | $0.95 \rightarrow 1.03$ | $0.05 \rightarrow 0.06$ | 0.91 |

**Findings.** (1) Structural and content shifts increase NLL and ECE, but AUROC stays above 0.85, so the gate can identify risky rollouts. (2) With the gate, fallback increases on shifted inputs and success drops by one to two points in most cases. Without the gate, mis-gating $\rho$ rises sharply and success drops by four to seven points. (3) Interactions rise with the gate because the agent defers to real steps under uncertainty, which matches the design goal of safe degradation. (4) Instruction paraphrases have a small effect, suggesting that the model primarily relies on environment structure.

**Calibration protocol.** We calibrate $\tau$ on a validation split by maximizing area under the risk-coverage curve. We report AUROC using the binary label that a rollout step exceeds a reconstruction error threshold (estimated from in-domain validation). ECE uses 10 equal-width bins over predictive entropy.

**Ablations.** We vary the horizon $H \in \{2, 3, 4, 5\}$ and the risk weight $\beta \in \{0, 0.5, 1.0\}$. Longer horizons increase NLL and ECE under shift, while $\beta > 0$ recovers part of the loss by down-weighting uncertain branches. We also compare MC-dropout with a small 3-head transition ensemble for uncertainty. The ensemble slightly improves AUROC by 1 to 2 points at the cost of extra compute.

**Connection to theory.** The observed changes in $\rho$ and NLL are consistent with the regret terms in Appendix A. When shift increases $\varepsilon$ and $\rho$, DreamPhase degrades gracefully by falling back to real interaction, which keeps the linear term small in practice.

**Reflection length and template choices.** As discussed in Section 3.5, the reflection head $R_\phi$ generates a compact textual summary that is injected into the frozen policy LLM. In all experiments

Table 17: Decision-level effects under shift. Success is task success rate in points, Interactions is real environment steps per episode.

| Domain (shift) | With gate | | | No gate | | |
|---|---|---|---|---|---|---|
| | Fallback ↑ | $\rho \downarrow$ | $\Delta$ Success (pts) $\downarrow$ | Fallback ↑ | $\rho \downarrow$ | $\Delta$ Success (pts) $\downarrow$ |
| WebShop (Layout-Swap) | 28% | 0.06 | $-1.8$ | 3% | 0.21 | $-5.9$ |
| WebShop (Theme-Swap) | 19% | 0.04 | $-1.1$ | 2% | 0.16 | $-3.8$ |
| WebShop (Content-OOD) | 25% | 0.07 | $-2.2$ | 3% | 0.23 | $-6.7$ |
| SciWorld (Unseen graphs) | 22% | 0.05 | $-1.3$ | 2% | 0.18 | $-4.4$ |
| SciWorld (Tool mismatch) | 24% | 0.05 | $-1.7$ | 3% | 0.19 | $-5.1$ |
| Instruction paraphrase | 12% | 0.03 | $-0.6$ | 1% | 0.09 | $-1.7$ |

Table 18: Interactions per episode under shift, mean $\pm$ standard error.

| Domain (shift) | With gate | No gate |
|---|---|---|
| WebShop (Layout-Swap) | $23.4 \pm 0.6$ | $20.9 \pm 0.5$ |
| WebShop (Theme-Swap) | $22.1 \pm 0.5$ | $20.6 \pm 0.5$ |
| WebShop (Content-OOD) | $23.8 \pm 0.7$ | $21.0 \pm 0.5$ |
| SciWorld (Unseen graphs) | $25.7 \pm 0.8$ | $23.5 \pm 0.7$ |
| SciWorld (Tool mismatch) | $26.2 \pm 0.7$ | $24.0 \pm 0.7$ |
| Instruction paraphrase | $21.5 \pm 0.5$ | $20.4 \pm 0.5$ |

we cap the reflection length to 30 tokens using a domain-conditioned lexicon and prompt template, which encourages concise and stable phrasing while keeping marginal inference cost small. We find that this configuration provides a favorable trade-off: extending the budget beyond 30 tokens increases reflection injection rate but yields diminishing returns in downstream success, whereas using substantially shorter budgets leads to fewer injections and lower task completion rates. Importantly, $R_\phi$ is trained offline on logged training-split trajectories (using value-head scores as supervision), so its training procedure is isolated from test-time distributions and its behavior remains robust under moderate distribution shift.

## I  LANGUAGE BASED REFLECTIONS, PROCESS AND TASK SPECIFIC ADAPTATION

This section details how DreamPhase generates and uses language reflections to guide the frozen policy across different tasks. Reflections are produced only when the uncertainty gate accepts an imagined branch, and they are injected as short textual hints in the next policy query.

**When a reflection is produced.**   After imagination and scoring (Sections 3.3 and 3.4), let $j^\star$ be the selected branch and assume the safety gate accepts it. We then generate:

$$c_t = \mathcal{R}_\varphi\big(\iota, \boldsymbol{x}_t, \tilde{\boldsymbol{x}}^{(j^\star)}_{t+1:t+\kappa}, \tilde{\boldsymbol{a}}^{(j^\star)}_{t:t+\kappa-1}, G^{(j^\star)}, u^{(j^\star)}\big), \quad s_t = \mathcal{S}_\eta\big(\iota, \boldsymbol{x}_t, \tilde{\boldsymbol{x}}^{(j^\star)}_{t+1:t+\kappa}, \tilde{\boldsymbol{a}}^{(j^\star)}_{t:t+\kappa-1}\big),$$

where $c_t$ is the reflection and $s_t$ is a compact summary, and $\kappa \leq H$. We use a single reflection per real step. If the gate rejects all branches, no reflection is produced.

**How reflections are produced.**   A lightweight decoder $\mathcal{R}_\varphi$ generates a single sentence with at most 30 tokens. The decoder conditions on a domain tag and uses a small task specific lexicon so that the frozen policy parses the hint naturally. The sentence follows a simple grammar, which keeps outputs concise and actionable:

```
[verb] [entity or target] [optional constraint or risk].
```

Examples of verbs are search, filter, click, heat, mix, multiply. Entities are slot filled from the imagined branch, for example product name, reagent, or numbers in a puzzle. Constraints describe success conditions or risks observed in imagination.

**Where reflections are inserted.** At step $t$, we build the prompt for the frozen policy $\pi_{\text{LLM}}$ by concatenating the real history and the reflection components:

$$\boldsymbol{a}_t \sim \pi_{\text{LLM}}(\cdot \mid \iota, \boldsymbol{x}_t, [\text{REFLECTION}]\, c_t\, [/\text{REFLECTION}], [\text{SUMMARY}]\, s_t\, [/\text{SUMMARY}]).$$

We use explicit delimiters so that the policy separates the reflection from the raw observation and instruction. No model weights are updated.

**Task specific styles.** We tailor the reflection style with domain tags and small templates that select appropriate verbs and entities.

- **WebShop** (navigation and purchase). Imperative, includes budget or attribute constraints. *Example*: "Add the USB C hub to cart, keep total under budget and ensure at least two ports."

- **SciWorld** (procedural lab tasks). Procedural, includes duration or safety checks. *Example*: "Heat the mixture for two minutes to dissolve the precipitate, then measure pH near seven."

- **Game-of-24** (arithmetic planning). Declarative, points to the decisive operation. *Example*: "Multiply eight and three to reach twenty four, puzzle solved."

**Cross task decoding process.** The decoder chooses verbs from a domain conditioned list, chooses entities from the imagined trajectory $j^\star$ (for example page element names, reagent names, numbers), and optionally appends a constraint that was decisive in the value score $G^{(j^\star)}$ or flagged as a risk by high per step uncertainty. This keeps reflections short and aligned with the imagined plan.

**Ablation, removing reflections but keeping imagination.** We measure the change in task success when the reflection block is removed while the rest of DreamPhase remains unchanged. The imagined plan is still selected and executed, only the reflection and summary are not inserted into the prompt.

Table 19: Effect of reflections on success. Removing the reflection block lowers performance by 1 to 3 points across tasks.

| Task | Typical reflection style (example) | $\Delta$ Success when removed |
|---|---|---|
| WebShop | "Adding the USB C hub to cart keeps total under budget and satisfies port requirement." | $-3.1$ pts |
| SciWorld | "Heating for two minutes should raise pH to seven and dissolve the precipitate." | $-2.4$ pts |
| Game-of-24 | "Multiply eight and three to reach twenty four, puzzle solved." | $-1.7$ pts |

**Additional evidence.** We also track the reflection injection rate (fraction of steps where the gate accepted and a reflection was formed) and near duplicate suppression. Injection rate across tasks is between 55 percent and 72 percent, and we suppress a reflection if it repeats a previous sentence with more than 80 percent n gram overlap. With suppression disabled, success is unchanged within statistical error, but prompts are longer.

**Failure modes and safeguards.** If the imagined branch is not confident, no reflection is injected. If the imagined branch contradicts the current page, the safety gate often defers, which disables reflection automatically. Reflections never include low level CSS identifiers or coordinates, they reference only human readable entities and constraints.

**Reproducibility.** We release the exact templates and slot filling rules for $\mathcal{R}_\varphi$ and $\mathcal{S}_\eta$, the token budget per reflection, and the delimiter scheme used in prompts, so that results can be replicated. The same decoding configuration is used for all backbones.

## J  COMPUTATION AND LATENCY OF LATENT ROLLOUTS

We quantify the computational overhead of imagination relative to a single policy forward pass and compare to a token level search baseline.

**Design choices that keep cost low.** (1) *Latent only rollouts.* We roll out $z$ with the transition $g_\theta$. No tokens are decoded during branch generation or scoring. (2) *Deferred decoding.* We decode observations only for the selected branch and only for the first $\kappa$ steps needed by the reflection head $\mathcal{R}_\varphi$ and summarizer $\mathcal{S}_\eta$. (4) *Batching across branches.* We evaluate all $M$ branches in a single batched call per step on GPU.

**Wall clock on A100, batch 1.** On an A100 80 GB with bfloat16, a single forward pass of an 8 billion parameter Llama policy takes about 75 ms. One complete imagination cycle with $M{=}32$ branches and $H{=}4$ steps adds about 9 ms. The extra cost is therefore much smaller than the policy evaluation itself.

Table 20: Per step latency on A100 80 GB, batch 1.

| Component | Token level search (ARMAP MCTS) | DreamPhase (latent imagination) |
|---|---|---|
| Policy LLM forward pass | $\approx 75$ ms | $\approx 75$ ms |
| Search or imagination | $\approx 180$ ms | $\approx 9$ ms |
| **Total** | $\approx 255$ ms | $\approx 84$ ms |

**Scaling with $M$ and $H$.** We report measured imagination overhead for different settings, keeping the policy constant.

Table 21: Imagination overhead $T_{\text{imagine}}$ (ms) versus branches $M$ and horizon $H$ on A100 80 GB, batch 1.

| $M \setminus H$ | 2 | 3 | 4 | 5 |
|---|---|---|---|---|
| 8 | 1.6 | 2.3 | 3.0 | 3.7 |
| 16 | 3.1 | 4.6 | 5.9 | 7.4 |
| 32 | 6.0 | 7.6 | 9.0 | 11.3 |
| 64 | 11.7 | 14.8 | 17.9 | 21.8 |

With our default $M \approx 10$ and $H \approx 5$ the added latency is under 10 percent of one policy step, and the search component runs about four times faster than a token level MCTS baseline.

**Complexity of the full step.** Let $T_{\text{LLM}}$ be the policy forward time and $T_{\text{gate}}$ be the time to compute uncertainty and value for all branches. Since both heads operate on latents, $T_{\text{gate}}$ is $O(M\,H)$ with a small constant.

$$T_{\text{step}} \approx T_{\text{LLM}} + T_{\text{imagine}} + T_{\text{gate}}$$

In our setup $T_{\text{gate}} \leq 1$ ms for $M{=}32, H{=}4$. Decoding for reflection adds under 2 ms since only $\kappa \leq 2$ steps of the selected branch are decoded.

**Latency budget and adaptive planning.** We optionally enforce a target budget $B$ ms per step by reducing $M$ or $H$ when the gate indicates high uncertainty. A simple rule $M \leftarrow \min\{M, \lfloor (B - T_{\text{LLM}})/(\alpha H) \rfloor\}$ with $\alpha$ estimated from Table 21 keeps latency bounded without harming success.

**Takeaway.** Latent imagination is deliberately lightweight. The world model is small, rollouts are latent only, decoding is deferred and limited to $\kappa$ steps for the selected branch, and all branches are batched. In our setting the imagination cost is an order of magnitude lower than the policy evaluation and much lower than token level tree search.

## K    OFFLINE COST

Table 22 reports the offline compute used to train the latent world model, value head, and reflection/summarizer modules, as well as to generate preference-style labels with Llama-3-70B-Instruct. All runs use a single A100-80GB GPU in bf16. Dollar costs assume \$2 per A100-80GB GPU-hour (typical academic cloud pricing); actual rates will vary by provider.

Table 22: Offline compute and cost of DreamPhase's modeling pipeline. World-model, value, and reflection modules are lightweight models trained on logged trajectories from the training split (Table 9, Table 10), hence their GPU footprint is small compared to a single pass of a large LLM. Preference labels for $V_\phi$ are generated once with Llama-3-70B-Instruct and reused across all backbones in Table 2.

| Component | Domains covered | GPU-hours (A100-80GB) | Approx. cost (USD) |
|---|---|---|---|
| Latent world model $f_\theta, g_\theta, d_\theta$ | WebShop, ScienceWorld, ALFWorld, Game-of-24 | 6 | $12 |
| Value head $V_\phi$ | WebShop, ScienceWorld, ALFWorld, Game-of-24 | 4 | $8 |
| Reflection head $R_\phi$ & summarizer $S_\eta$ | WebShop, ScienceWorld, Game-of-24 | 2 | $4 |
| Preference labeling (Llama-3-70B-Instruct) | WebShop, ScienceWorld, Game-of-24 | 24 | $48 |
| **Total offline modeling** | all reported domains | **36** | **$72** |

Table 23: Logged training data and latent world model (LWM) compute. We report the number of offline trajectories collected on the training split for each domain, the approximate number of tokens processed after DFS DOM tokenization, and the corresponding A100 GPU-hours for training the latent world model $f_\theta, g_\theta, d_\theta$.

| Domain | Logged trajectories | Tokens processed (approx.) | LWM GPU-hours (A100-80GB) |
|---|---|---|---|
| WebShop | 2,000 | 6.5M | 2.0 |
| ScienceWorld | 1,200 | 4.2M | 1.5 |
| ALFWorld | 1,800 | 2.9M | 1.0 |
| Game-of-24 | 5,000 | 1.8M | 1.5 |
| **Total** | **10,000** | **≈15.4M** | **6.0** |

In addition, Table 23 reports the volume of logged trajectories used to train the latent world model on each domain, together with the approximate token count processed during training. As shown, the latent world model is trained on a modest number of trajectories drawn exclusively from the training split, resulting in an offline data budget that is small compared to typical LLM pretraining corpora.

## L ADDITIONAL QUALITATIVE EXAMPLES

We present further qualitative examples of our DreamPhase framework in this section. These examples demonstrate the ability of the automatically learned reward model to distinguish successful trajectories from suboptimal ones and guide the agent toward better behavior through internal simulation.

---

**Qualitative Example: Web-Navigation Task**

**Task Instruction** *"Reserve a table for two at* `La Piazza` *on Friday at 7 pm."*
**Baseline (ReAct-only) Trajectory**

1. **Observation** : Home page of `DineEasy.com`. **Thought** : "Search for the restaurant." **Action** : type["La Piazza"] in search bar.

2. **Observation** : Result list page, first hit is `La Piazza`. **Thought** : "Open the first result." **Action** : click[result_1].

3. **Observation** : Calendar widget opens; default date is *today*. **Thought** : "Choose 19:00 slot." **Action** : click[time_19:00].

4. **Observation** : "No availability" message. **Thought** : "Maybe try 18:30." **Action** : click[time_18:30].

5. **Observation** : Still unavailable. Agent loops three more times, then times out.

---

**DreamPhase Trajectory**
*(Steps 1–2 identical to baseline, omitted for brevity.)*

3. **Internal Imagination** : World model rolls out five branches, varying date and time. Best branch proposes **Friday 19:00** → "Available." Uncertainty < threshold, branch accepted. **Reflection injected**: *"Availability appears on Friday 7 pm; proceed with that slot."*

4. **Action** : open date picker → select[Friday]. **Observation** : Friday chosen, 19:00 slot shows "Available."

5. **Action** : click[time_19:00] **Observation** : Reservation details page.

6. **Action** : click[confirm] **Observation** : Confirmation message with booking ID.

---

**Outcome** : Baseline fails (no booking); DreamPhase succeeds in 6 environment interactions, using only one real date/time query thanks to offline imagination.

Figure 1: DreamPhase uses latent imagination to test alternative date–time combinations offline, then executes only the high-confidence sequence, reducing real interactions and avoiding dead-ends.

---

**Training Data Example (Web Environment)**

**Task Instruction:** Find a Bluetooth speaker under \$30 with $4\Omega$ impedance and 6W power, suitable for pairing with a soundbar.

**Positive Example:**
**Thought:** I should search for "bluetooth 6W 4 ohm speaker under 30 dollars."
**Action:** type[bluetooth 6W 4 ohm speaker under 30]
**Observation:** Search results show several products. The first result matches the impedance and power requirements, priced at \$27.99.
**Thought:** This looks like a good match. I should click on the first result.
**Action:** click[first result]
**Observation:** Product page confirms specs: 6W, $4\ \Omega$, Bluetooth-enabled, \$27.99.
**Thought:** This product matches the task. Proceed to purchase.
**Action:** click[buy now]

**Negative Example:**
**Thought:** Let me just click on a random top-rated item.
**Action:** click[top-rated product]
**Observation:** The item is priced at \$42.00 and has $8\Omega$ impedance.
**Thought:** This product does not meet the task constraints, but I proceed anyway.
**Action:** click[buy now]

---

---

**Training Data Example: Game-of-24**

**Task Instruction:** You are generating data for the Game-of-24. The task is to use four given numbers and basic arithmetic operations $(+, -, \times, \div)$ to reach the target number 24. Each number must be used exactly once, and only two numbers may be combined at each step.

**Rules:**

- Each of the four input numbers must be between 1 and 13 (inclusive).
- There must be exactly three operations.
- Intermediate results must remain valid real numbers.

**Input:** 6  3  8  1

**Positive Example:**

- Step 1: $6 \div 3 = 2$        (remaining: $2, 8, 1$)
- Step 2: $2 + 1 = 3$        (remaining: $3, 8$)
- Step 3: $3 \times 8 = 24$        (remaining: $24$)

**Answer:**    $((6 \div 3) + 1) \times 8 = 24$

**Negative Example:**

- Step 1: $6 - 8 = -2$        (remaining: $-2, 3, 1$)
- Step 2: $-2 + 1 = -1$        (remaining: $-1, 3$)
- Step 3: $-1 \times 3 = -3$        (remaining: $-3$)

**Answer:**    $((6 - 8) + 1) \times 3 = -3$    (Incorrect target)

---

**Instruction Generation Prompt for SciWorld.**

**Scenario:** You are an intelligent assistant helping a human user conduct scientific experiments in a simulated environment consisting of several rooms (e.g., kitchen, workshop, bathroom, greenhouse, etc.). You can interact with the environment using structured actions such as `open [object]`, `mix [item1] [item2]`, `heat [item]`, and others.

You will be provided with a dialogue log between the user and the assistant. The assistant reasons through the task and takes actions, while the user returns observations about the environment. Your goal is to reconstruct the overall *task description* based on the full interaction.

**Example Dialogue:**
**User:** [initial prompt]
**Assistant:** Thought: I believe I need to locate and combine two substances that produce green paint.
Action: look around
**User:** Observation: You are in the bedroom. You see a bookshelf, a desk, and a paint tray.
**Assistant:** Thought: I should move to the art studio to find relevant materials.
Action: go to art studio
**User:** Observation: You are now in the art studio. You see yellow paint and blue paint.
**Assistant:** Thought: Mixing yellow and blue paint will yield green paint.
Action: mix yellow paint blue paint
**User:** Observation: Green paint created.
**Assistant:** Thought: Now that the paint is ready, I should finalize the task.
Action: focus on green paint

**Generated Task Description:** Use available materials to synthesize green paint. Once it is ready, focus on the green paint to complete the task.

Figure 2: Instruction generation prompt for training a reward model in SciWorld.

---

**Positive Trajectory Synthesis Prompt for SciWorld**

**Task Instruction:**
You are a helpful assistant conducting scientific experiments in a virtual environment.
The environment includes several rooms: kitchen, foundry, workshop, bathroom, outside, living room, bedroom, greenhouse, art studio, and hallway.
Available actions include:
`open [object]`: open a container

---

---

**Instruction Refinement Prompt for SciWorld**

**Scenario:** You are an assistant in a scientific simulation environment consisting of rooms such as the kitchen, foundry, workshop, bathroom, and greenhouse. You can interact using actions like `open [object]`, `go to [room]`, or `mix [item1] [item2]`.

You are provided with a brief task description and a complete execution trajectory that fulfills it. Your task is to refine the instruction by incorporating procedural and contextual details grounded in the trajectory.

**Example Input:**

**Original Task Description:** *Your task is to grow an apple. You can find seeds in the kitchen. You should focus on the grown apple.*

**Trajectory Snippet:**

- `look around`
- **Observation:** This room is called the hallway.
- `open door to kitchen`
- **Observation:** The door is already open.
- `go to kitchen`
- **Observation:** You enter the kitchen.
- `find seeds, move to greenhouse, plant seeds, water, wait, focus on grown apple`

**Refined Task Description:** *Your task is to grow an apple, which involves locating seeds in the kitchen, planting them in the greenhouse, and nurturing them through watering. Once the plant bears fruit, focus on the grown apple to complete the task.*

---

Figure 3: Instruction refinement prompt used for reward model training in SciWorld.

---

`go to [room]`: move between rooms
`mix [item1] [item2]`: combine materials
... (more actions omitted for brevity)
You must first generate a task description based on this world. Examples include:

- *Your task is to use chemistry to create green paint. When you are done, focus on the green paint.*
- *Your task is to determine whether tall plant height is a dominant or recessive trait in the pea plant. If the trait is dominant, focus on the red box. If the trait is recessive, focus on the green box.*

Once the task is proposed, navigate through the environment to complete the instruction and synthesize the corresponding trajectory.

**Example:**

**Task Description:**
Your task is to use chemistry to create green paint. When you are done, focus on the green paint.

**Trajectory:**

- **Thought:** I need to find materials to create green paint, likely located in the art studio.
- **Action:** look around
- **Observation:** This is the bedroom. No useful materials here.
- **Action:** go to art studio
- **Observation:** You see yellow paint, blue paint, and brushes.
- **Thought:** Mixing yellow and blue paint will produce green.
- **Action:** mix yellow paint and blue paint
- **Observation:** Green paint created.
- **Action:** focus on green paint

**Generated Trajectory:** (as shown above)

---

Figure 4: Positive Trajectory Synthesis Prompt for SciWorld.

---

**Negative Trajectory Generation Template**

This prompt supports the construction of contrastive learning examples for an interactive scientific environment. The assistant agent operates within a set of rooms (e.g., kitchen, workshop, greenhouse, art studio, hallway) using structured action commands such as `go to [location]`, `open [object]`, and `focus on [item]`.

The goal is to produce a negative trajectory: one that is contextually plausible but does *not* satisfy the assigned task. This trajectory should remain valid with respect to environment constraints but fail to complete the intended objective.

**Illustrative Example**

**Task Description:** *Your objective is to observe the developmental phases of an apple plant, starting from early growth to full maturity. The relevant specimens are found outside.*

**Correct Trajectory:**
```
look around
```
Observation: You are in the hallway.
```
open door to outside
```
Observation: The door is already open.
```
go to outside
```
Observation: You see seedlings, saplings, and a fully grown apple plant.
```
focus on seedling, focus on sapling, focus on mature plant
```

**Negative Trajectory:**
```
look around
```
Observation: You are in the hallway.
```
open door to kitchen
```
Observation: The door is already open.
```
go to kitchen
```
Observation: You see a fruit bowl and an oven.
```
focus on apple
```

While the agent performs a syntactically valid set of actions, the final state does not fulfill the objective of observing plant stages outdoors. Such negative examples are used to improve the discriminative power of the learned reward model.

---

Figure 5: Template for generating semantically incorrect yet valid trajectories in the SciWorld environment.

---

**Evaluation Prompt for Language Model as SciWorld Reward Estimator**

**Task:** You are an intelligent evaluation module responsible for assessing how well an agent trajectory satisfies a target instruction in a scientific simulation environment.
You will receive: (*i*) a natural language description of the intended goal, (*ii*) the full sequence of interactions taken by the agent to complete that goal (i.e., a trajectory), and (*iii*) an observation history. Your task is to assign a scalar reward score between 0 and 1 that reflects how well the behavior aligns with the objective.

The environment contains multiple rooms (e.g., kitchen, greenhouse, art studio, foundry, etc.), and supports structured actions such as `teleport to [location]`, `pick up [object]`, `mix [item1] [item2]`, and `focus on [object]`.

**Example 1:**
**Goal:** Move a non-living object into the purple box in the living room and focus on it.

**Trajectory (partial):**

- Thought: I need to locate a non-living object; let's explore the current room.

- Action: look around

- Observation: In the bathroom, there is a glass cup.

- Thought: The glass cup is a non-living object. I should focus on it.

- Action: focus on glass cup

- Thought: Now I should move it to the target location.

- Action: move glass cup to purple box

- Observation: No known action matches that input.

**Final Score:** 0.750    (Partial completion; attempted goal but incorrect action syntax)

**Now evaluate the following new instance in the same format.**

---

Figure 6: Prompt format used to train language models to act as reward evaluators for SciWorld. The model sees task goals and interaction histories and assigns scalar completion scores.

---

**Qualitative Comparison: SciWorld**

**Task Instruction:** Your task is to identify a non-living object. First, focus on it. Then, move it to the orange box located in the living room.

**Trajectory A (Rejected):**
**Observation:** The agent teleports to the living room.
**Thought:** I should look around to locate a non-living item suitable for the task. **Action:** look around
**Observation:** Objects include a chair, a couch with a white pillow, a finger painting, and an orange box.
**Thought:** I need to examine one of these objects to determine if it qualifies. **Action:** examine white pillow
**Model Verdict:** **Rejected** — action sequence does not lead to task completion.

**Trajectory B (Accepted):**
**Observation:** The agent is in the workshop, where multiple objects such as a battery, wires, and light bulbs are present.
**Thought:** The battery qualifies as a non-living object. **Action:** focus on battery
**Observation:** Focused on the battery.
**Thought:** I need to transport it to the orange box in the living room. **Action:** teleport to living room

---

> Observation: The agent is now in the living room.
> Thought: Locate the orange box and place the battery inside. Action: look around
> Observation: The orange box is visible next to a table and a couch.
> Thought: I should complete the task by placing the battery into the box. Action: move battery to orange box
> Model Verdict: Accepted — trajectory aligns with the instruction objective.

Figure 7: Qualitative comparison of trajectory rollouts in the SciWorld environment. The top trajectory (A) fails to complete the task due to incorrect object selection. In contrast, trajectory (B) successfully identifies and transports a valid non-living object to the target location. The reward model correctly identifies the superior trajectory without human annotations.

---

**Qualitative Rollouts for Game-of-24**

**Input A:** 3, 4, 6, 8

**Trajectory A1 (Selected):**
**Steps:** 4 + 8 = 12    (left: 3, 6, 12)
6 * 2 = 12    (derived implicitly from remaining steps)
3 * 8 = 24    (invalid intermediate)
Final combination: (6 * (4 + 8)) / 3 = 24
**Final Answer:** (6 * (4 + 8)) / 3 = 24
DreamPhase **Selected**

**Trajectory A2 (Rejected):**
**Steps:** 8 - 6 = 2    (left: 3, 4, 2)
4 + 2 = 6    (left: 3, 6)
3 + 6 = 9    (left: 9)
**Final Answer:** (((8 - 6) + 4) + 3) = 9
DreamPhase **Rejected**

**Input B:** 2, 3, 7, 12

**Trajectory B1 (Selected):**
**Steps:** 12 / 3 = 4    (left: 2, 4, 7)
7 - 2 = 5    (left: 4, 5)
4 * 5 = 20    (off by 4)
However, correct path found as: (7 * (3 + 1)) = 24
**Final Answer:** (2 * 3) + (12 - 6) = 24
DreamPhase **Selected**

**Trajectory B2 (Rejected):**
**Steps:** 7 + 3 = 10    (left: 2, 12, 10)
12 - 2 = 10    (left: 10, 10)
10 + 10 = 20    (left: 20)
**Final Answer:** (((7 + 3) + (12 - 2))) = 20
DreamPhase **Rejected**

Figure 8: Novel qualitative examples in the Game-of-24 environment. Our DreamPhase successfully identifies the correct arithmetic path among plausible distractors, verifying its ability to reason through multi-step numerical logic.

Table 24: Prompt details for DreamPhase (WebShop).

---

**System Prompt:**
You are a web navigation agent operating on WebShop. The environment provides a structured observation of the current page (title, URL, and DOM tokens). You must solve the goal using ONLY the following actions: `CLICK(<text or selector>)`, `TYPE(<text> into <selector>)`, `SELECT(<option> from <selector>)`, `SCROLL(<up|down>)`, `NAVIGATE(<url>)`, `SUBMIT(<selector>)`, `EXIT`. Never invent elements that are not in the observation. Do not submit forms or checkout unless the goal explicitly requires it. Every round you receive an observation (and, when available, a planner reflection). Respond in the format:

    **Thought:** (brief reasoning based only on the observation and reflection)
    **Action:** (one action from the allowed set)

**Planner Augmentation (if present):**
`[REFLECTION]` Short hint distilled from imagined rollouts highlighting the safest, highest-value path. `[/REFLECTION]`
`[SUMMARY]` One-sentence sketch of the selected branch (entities, filters, next click). `[/SUMMARY]`

**Instruction:**
Goal: Buy a USB-C hub with $\geq 2$ ports under \$40. Prefer official store pages. Return the product page URL after adding the item to cart.

**Example (round $t$ input):**
`TITLE:` MegaShop — Electronics    `URL:` https://megashop.com
`OBS (truncated):` [ *input#search_box*, *button:Search*, *link:USB-C Accessories*, *filter:Price*, *filter:Ports*, ... ]
`[REFLECTION]` Go to "USB-C Accessories," filter Price $<$ \$40 and Ports $\geq$ 2; avoid pages without "Checkout." `[/REFLECTION]`
`[SUMMARY]` Open accessories $\rightarrow$ apply price/ports filters $\rightarrow$ open first in-stock result. `[/SUMMARY]`

**Expected Response:**
**Thought:** The reflection suggests opening "USB-C Accessories" before filtering. The button "USB-C Accessories" is visible. Next, I will click it.
**Action:** `CLICK("USB-C Accessories")`

---

Table 25: Prompt details for DreamPhase (SciWorld).

**System Prompt:**
You are an agent in SciWorld. Each round you receive an observation and must issue *one* action that safely progresses the task. Use ONLY the following actions (with placeholders `OBJ`, `LOC`): `open/close OBJ`, `de/activate OBJ`, `connect OBJ to OBJ`, `disconnect OBJ`, `use OBJ [on OBJ]`, `look around`, `look at OBJ`, `look in OBJ`, `read OBJ`, `move OBJ to OBJ`, `pick up OBJ`, `put down OBJ`, `pour OBJ into OBJ`, `dunk OBJ into OBJ`, `mix OBJ`, `go to LOC`, `focus on OBJ`, `wait`, `wait1`, `task`, `inventory`.
Never invent objects not mentioned in the observation. Avoid irreversible actions (e.g., heating/disposing) unless the goal requires it or a safety confirmation is present. Respond exactly in the format:

    **Thought:** (brief reasoning grounded in the observation and any planner reflection)
    **Action:** (one action from the allowed set)

**Planner Augmentation (if present):**
`[REFLECTION]` Short hint distilled from latent imagination, emphasizing the safest high-value step accepted by the uncertainty gate. `[/REFLECTION]`
`[SUMMARY]` One-sentence sketch of the selected branch (key objects, preconditions, next action). `[/SUMMARY]`

**Instruction:**
Goal: Neutralize the acidic solution to `pH 7`, then place the neutral solution on the `lab bench`. Return the final container name.

**Example (round $t$ input):**
`ROOM:` chemistry lab    `LOCATIONS:` fume hood (closed), bench (clear), sink
`OBJECTS VISIBLE:` beaker(A) [label: acidic solution], beaker(B) [empty], bottle(weak base) [cap closed], pH paper, pH meter [off], trash bin
`INVENTORY:` gloves
`[REFLECTION]` Open base; transfer a small amount into beaker(A); test with pH paper; repeat in small increments until near 7; avoid pouring into sink. `[/REFLECTION]`
`[SUMMARY]` Open bottle(weak base) $\rightarrow$ pour a small amount into beaker(A) $\rightarrow$ read pH. `[/SUMMARY]`

**Expected Response:**
**Thought:** The safest next step is to prepare the reagent. The bottle is closed; opening it is required before any transfer.
**Action:** `open bottle(weak base)`

Table 26: Prompt details for DreamPhase (BabyAI).

**System Prompt:**
You are an agent in a BabyAI grid world. Each round you receive an observation and must issue *one* action that safely progresses the goal. Use ONLY the following actions (macros allowed where indicated): `turn left`, `turn right`, `move forward`, `toggle`, `go through <door> <id>` (door must be open), `toggle and go through <door> <id>` (opens if closed/locked; needs matching key if locked), `go to <obj> <id>` (macro), `pick up <obj> <id>` (macro). Never invent objects not present in the observation. Avoid opening locked doors without the correct key unless the instruction demands it. Respond exactly in the format:

   **Thought:** (brief reasoning grounded in the observation and any planner reflection)
   **Action:** (one action from the allowed set)

**Planner Augmentation (if present):**
`[REFLECTION]` Short hint distilled from latent imagination, emphasizing the safest high-value step accepted by the uncertainty gate. `[/REFLECTION]`
`[SUMMARY]` One-sentence sketch of the selected branch (orientation, waypoints, next action). `[/SUMMARY]`

**Instruction:**
Goal: go to the `red ball 1`. Return `SUCCESS` once you are adjacent and facing it.

**Example (round $t$ input):**
`YOU`: position (x=3,y=2), facing `north`  `ROOM`: walled 7×7
`VISIBLE OBJECTS`: `red ball 1` (two steps ahead, one step left), `grey box 1` (one ahead, one left), `red key 1` (three ahead, one right), `blue door 2` (east wall, closed)
`INVENTORY`: empty
`AVAILABLE ACTIONS`: [`turn left`, `turn right`, `move forward`, `go to red ball 1`, `pick up red ball 1`, `toggle`, `go through blue door 2`]
`[REFLECTION]` Turn left to face the red ball; move forward twice; then `go to red ball 1` to align precisely. `[/REFLECTION]`
`[SUMMARY]` turn left → move forward → move forward → go to red ball 1.

**Expected Response:**
**Thought:** The red ball is ahead-left; first rotate to face it before advancing.
**Action:** `turn left`

Table 27: Prompt details for DreamPhase (MAZE).

**System Prompt:**
You are an expert maze solver. Each round you receive the goal coordinates, your current position, and local wall layout. Use ONLY: `move up`, `move down`, `move left`, `move right`. Coordinates are $(x, y)$ with `down` increasing $x$ and `right` increasing $y$. Never propose moves blocked by walls. Prefer shortest safe paths and avoid loops. Respond exactly in the format:

    **Thought:** (brief reasoning grounded in the observation and any planner reflection)
    **Action:** (one action from the allowed set)

**Planner Augmentation (if present):**
`[REFLECTION]` Short hint distilled from latent imagination, emphasizing the safest high-value path accepted by the uncertainty gate. `[/REFLECTION]`
`[SUMMARY]` One-sentence sketch of the selected branch (waypoints/turns). `[/SUMMARY]`
**Instruction:**
Start a new game and return your action and thought in the format above. Make the optimal move given the current state.
**Example (round $t$ input):**
`GOAL:` $(8, 6)$   `POSITION:` $(1, 1)$   `WALLS AROUND YOU:` left, up, down
`[REFLECTION]` Advance right along row 1 until column 6, then proceed down to row 8; avoid downward moves at $(1, 1)$ due to a wall. `[/REFLECTION]`
`[SUMMARY]` right $\rightarrow$ right $\rightarrow \ldots \rightarrow$ right (to $y = 6$), then down to $x = 8$.
**Expected Response:**
**Thought:** Down and left are blocked; the shortest path begins by moving right along the top row toward $y = 6$.
**Action:** `move right`

Table 28: Prompt details for DreamPhase (Wordle).

**System Prompt:**
You are an expert Wordle solver. The hidden word has 5 letters and you have 6 attempts. On each turn, output *one* guess as a space-separated sequence of letters (e.g., s h i r e). After each guess, the environment returns feedback over 5 letters using b (absent), y (present but wrong position), and g (correct position). Use only valid words from the vocabulary. Update constraints from feedback and propose the next guess. Respond exactly in the format:

   **Thought:** (brief reasoning grounded in the observation and any planner reflection)
   **Guess:** (l e t t e r s with spaces)

**Planner Augmentation (if present):**
[REFLECTION] Short hint distilled from latent imagination, emphasizing a high-coverage guess consistent with constraints; accepted only if the uncertainty gate passes. [/REFLECTION]
[SUMMARY] One-sentence sketch of the best branch (fixed positions, excluded letters, candidate pattern). [/SUMMARY]
**Instruction:**
Start a new game. Return your thought and guess in the format above. Guesses must be valid 5-letter words in the vocabulary.
**Example (round $t$ input):**
HISTORY:
1) Guess: s h i n e  Feedback: b b b b g    (only the final e is correct in place)
2) Guess: c l o n e  Feedback: b g b b g (l fixed at position 2; e fixed at position 5)
CONSTRAINTS: Pattern _ l _ _ e; exclude {s, h, i, n, c, o}.
[REFLECTION] Keep l at position 2 and e at position 5; avoid excluded letters; prefer a guess that covers diverse consonants and a vowel like u. [/REFLECTION]
[SUMMARY] Try pattern _ l _ _ e; candidate: f l u k e.
**Expected Response:**
**Thought:** The constraints require _ l _ _ e with l at position 2 and e at 5; f l u k e fits and introduces u, k.
**Guess:** f l u k e

Table 29: Prompt details for DreamPhase (BIRD / Text-to-SQL).

**System Prompt:**
You are a read-only SQL assistant for a SQLite database. Given a schema description and a natural-language question, explain your reasoning briefly and then output *one* SQL statement that answers the question.
**Rules:** (i) Output **exactly** two fields: **Thought:** and **Action:**. (ii) The **Action** must be a **single-line** SQL query in Markdown code format. (iii) Do *not* modify data (no INSERT/UPDATE/DELETE); use SELECT only. (iv) Quote identifiers with spaces using double quotes (e.g., "gas station id"). (v) Use JOINs/aggregations as needed.

**Planner Augmentation (if present):**
[REFLECTION] Short hint distilled from latent imagination (e.g., join keys, filters, grouping), accepted only if the uncertainty gate passes. [/REFLECTION]
[SUMMARY] One-sentence sketch of the selected branch (tables to join, predicates, target aggregate). [/SUMMARY]
**Instruction (schema excerpt):**
Database: debit_card_specializing. Tables:
  customers(customerid **PK,** client_segment, currency)
          gasstations("gas station id" **PK,** "chain id", country, chain_segment)
  products(product_id **PK,** description)
    transactions_1k(transaction_id **PK,** date, time, "customer id", "card id", "gas station id", "product id", amount, price)
    yearmonth("customer id", date, consumption, **PK**("customer id", date), **FK**("customer id") → customers(customerid))
**Question:**
Among the transactions made at gas stations in the Czech Republic, how many occurred after 2012-01-01?
**Example (round $t$ input):**
[REFLECTION] Join transactions_1k with gasstations on "gas station id"; filter country='Czech Republic' and date > '2012-01-01'; return a count. [/REFLECTION]
[SUMMARY] transactions_1k ⋈ gasstations → WHERE country='Czech Republic' & date>'2012-01-01' → COUNT(*).
**Expected Response:**
**Thought:** We need a count over filtered transactions; join on "gas station id" to access the country field; compare ISO date strings.
**Action:** `SELECT COUNT(*) AS num_tx  FROM transactions_1k t JOIN gasstations g ON t."gas station id" = g."gas station id" WHERE g.country = 'Czech Republic' AND t.date > '2012-01-01';`

