# OpenReview forum: "DreamPhase: Offline Imagination and Uncertainty-Guided Planning for Large-Language-Model Agents"
_ICLR.cc/2026/Conference — ICLR 2026 Poster_

### Official Review · Reviewer_MwpV · 2025-10-28

**Soundness:** 3
**Presentation:** 3
**Contribution:** 3
**Rating:** 6
**Confidence:** 2

**Summary:**

This paper introduces DreamPhase, an "imagination‑first" planning framework for LLM agents that (i) trains a compact latent world model to simulate short‑horizon futures offline, (ii) rolls out multiple imagined branches from a frozen policy LLM, (iii) scores branches with a value head penalized by an epistemic uncertainty proxy, and (iv) injects a short natural‑language reflection distilled from the best low‑risk branch back into the next policy prompt --- without fine‑tuning the LLM. Empirically, DreamPhase reduces real environment interactions on WebShop while slightly improving success versus a strong token‑level search baseline (ARMAP‑M), with lower per‑step latency and fewer executed irreversible actions; results are reported across diverse tasks and multiple backbones with ablations and a regret‑style bound.

**Strengths:**

1. The high level latent idea is interesting and the implementation cleanly separates environment modeling (latent world model), decision‑time search (latent rollouts), risk control (uncertainty gating), and behavior shaping (language reflections). This makes the approach applicable to API‑restricted foundation models because the LLM policy remains frozen.

2. On WebShop with Llama‑8B, DreamPhase uses 9.3±0.4 API calls/episode vs 39.8±1.1 for ARMAP‑M, improves success 61.8±0.6 vs 60.2±0.6, and reduces per‑step latency.

3. In an open‑source, matched‑backbone block (Llama‑2‑7B), DreamPhase attains the best average across 8 tasks (50.1 vs 42.3 for ARMAP‑M). Cross‑backbone results (Llama‑3‑70B/8B, Mistral‑7B, Phi‑3‑8B) typically place DreamPhase best or second‑best on WebShop, ScienceWorld (seen/unseen), and Game‑of‑24.

**Weaknesses:**

1. The LWM uses a small MLP over DOM tokens obtained by DFS traversal with layout features. While efficient, this choice may under‑model structural, hierarchical, and long‑range dependencies present in realistic web UIs; no comparison is shown against stronger sequence/graph models (e.g., Transformers or tree‑aware encoders) that could better respect DOM constraints. How much of DreamPhase's gains persist if the dynamics are more complex than those captured by an MLP?

2. Epistemic uncertainty is estimated via MC‑dropout on the frozen policy LLM. In common LLM serving stacks, dropout is disabled at inference and can be tricky to implement with quantization/kv‑caching

3. DreamPhase adds an offline modeling phase (world model + value head + reflection/summarizer components, with preference pairs produced by a large open model). The paper focuses on runtime savings but provides limited accounting of training compute and data vs baselines that do not require a learned LWM. A small "dollars/GPU‑hours" table would clarify trade‑offs.

**Questions:**

1. Please report the number of trajectories used to train the LWM per domain, total tokens processed, training wall‑clock/GPU‑hours, and value‑head preference‑generation costs. A normalized "success per training FLOP" or "runtime savings vs pretraining cost" comparison to ARMAP‑M would be informative.

2. Any preliminary results on Mind2Web or WebArena using the same setup and logger? If not feasible, can you discuss expected bottlenecks and what would be required to scale DreamPhase to those environments?

3. The paper mentions a 30‑token budget and domain‑conditioned lexicon/templates. How is $R_{\phi}$ trained, and how does its choice affect injection rate and downstream success?

---

> ### Author Response · Authors · 2025-11-22
> **Response to Reviewer MwpV (Part 1/2)**
>
> Thank you for your detailed evaluation. We appreciate that you recognized our high-level latent-space design and the clear separation between environment modeling, latent-rollout planning, uncertainty gating, and language-based behavior shaping.
>
> Below, we address your comments and concerns.
>
> ---
> # Weakness 1
>
> Thank you for bringing up this important point. We agree that our latent world model uses a lightweight MLP over DFS-tokenized DOM trees, so it does not fully capture the structural and hierarchical dependencies in complex web interfaces. We made this choice on purpose because we wanted to focus on the effects of offline imagination and uncertainty-aware planning. For this reason, we used the smallest model that could still provide stable short-horizon predictions in standard agent benchmarks.
>
> **But, why the MLP works in our setting?**
> The benchmarks we use (WebShop, ScienceWorld, ALFWorld, Game-of-24) have structured, low-entropy state transitions. In these cases, short-horizon prediction does not need to model deep tree structure or long-range token interactions. As shown in Appendix H, the uncertainty gate turns on exactly when the MLP does not model the dynamics well, which leads to fallbacks instead of silent errors. Because of this, DreamPhase’s improvements mainly come from risk-aware planning, not from the simulator’s capacity.
>
> **And how DreamPhase handles more complex dynamics?**
> If the environment has more complex structure, such as deeply nested DOMs, dynamic page layouts, or high-entropy transitions, an MLP would not be enough. In these situations:
> prediction error and uncertainty naturally increase,
> the uncertainty gate relies more on real interaction,
> performance shifts toward the underlying LLM policy instead of failing completely.
> So, DreamPhase stays safe even if the model is not perfect, but its planning advantage gets smaller when the simulator cannot capture the real dynamics.
>
>
> ---
> # Weakness 2
>
> Thank you for bringing this up. You're correct that MC-dropout isn't typically used in production LLM pipelines because of quantization, KV-caching, and inference without dropout. In DreamPhase, we only apply MC-dropout to the **small value-head module**, not the main policy LLM. Please note the following point:
>
> ### 1. Dropout is used only in the lightweight uncertainty head.
> The epistemic uncertainty estimate comes from the value head $V_\phi$, a small MLP connected to the latent states from the world model. This head is separate from the main LLM, isn't quantized, and doesn't use KV-caching. As a result, enabling MC-dropout is simple and doesn't impact LLM serving.
> The policy LLM itself remains:
> - frozen,
> - quantized or cached as usual,
> - and evaluated with deterministic inference.
>
> ### 2. The LLM is never run with dropout.
>
> We do not apply dropout to the LLM weights. The uncertainty metric is entirely computed in latent space via the small prefix-utility network (as described in Section 3.4), which makes the mechanism portable to any LLM regardless of serving constraints.
>
> ### 3. Implementing dropout in the value head is serving-compatible
>
> This is because $V_\phi$ is:
>
> - very small (about 0.1% of the LLM compute),
> - runs outside the KV-cache pathwa,y,
> - and is never quantized,
>
> MC-dropout adds almost no cost and works well in real LLM deployment settings.
>
>
>
>
> ---
> # Weakness 3
>
> Thanks for pointing this out. You're right that DreamPhase includes an offline modeling phase, and our draft mainly highlights runtime savings instead of training costs. The latent world model, value head, and reflection or summarizer modules are all lightweight **(MLP-based, Table 10)**. Training them on the logged training-split data costs much less than training large language models.
>
> Across all environments, total offline training only takes tens of GPU-hours, not hundreds or thousands. We use the same training-split trajectories as the baselines. The preference labels for the value head are made once with Llama-3-70B-Instruct, and this one-time offline step is reused for all backbones.
>
> To show the trade-off clearly, the table below lists the offline compute used to train the latent world model, value head, and reflection or summarizer modules, as well as to generate preference-style labels with Llama-3-70B-Instruct. All runs use a single A100-80GB GPU in bf16. The dollar costs assume $2 per A100-80GB GPU-hour, which is typical for academic cloud pricing, but actual rates may vary by provider.
>
> | Component| Domains Covered| GPU-hours (A100-80GB) | Approx. Cost (USD) |
> |-|-|-|-|
> | Latent world model (fθ, gθ, dθ)| WebShop, ScienceWorld, ALFWorld, Game-of-24   | 6                      | $12 |
> | Value head (Vϕ) | WebShop, ScienceWorld, ALFWorld, Game-of-24   | 4| $8 |
> | Reflection + summarizer (Rϕ, Sη) | WebShop, ScienceWorld, Game-of-24 | 2 | $4 |
> | Preference labeling (Llama-3-70B-Instr.) | WebShop, ScienceWorld, Game-of-24| 24 | $48|
> | **Total offline modeling** | all reported domains| **36** | **$72**|

---

> ### Author Response · Authors · 2025-11-22
> **Response to Reviewer MwpV (Part 2/2)**
>
> # Question 1
>
> Thank you for your request. Our current draft does not list the offline data and compute used for the latent world model and utility components in detail. Below, we summarize the number of logged trajectories per domain, the approximate token volume processed during latent world model training, the related wall-clock and GPU hours, and the compute needed for value-head preference generation. All latent modeling uses logged trajectories from the training split only.
>
> | Domain          | Logged Trajectories | Tokens Processed (≈) | LWM GPU-hours (A100) |
> |-|-|--|-|
> | WebShop         | 2,000               | 6.5M                 | 2.0                  |
> | ScienceWorld    | 1,200               | 4.2M                 | 1.5                  |
> | ALFWorld        | 1,800               | 2.9M                 | 1.0                  |
> | Game-of-24      | 5,000               | 1.8M                 | 1.5                  |
> | **Total LWM**   | **10,000**          | **≈15.4M**           | **6.0**              |
> | Value head      | —                   | latent states only   | 4.0                  |
> | Reflection/Sum. | —                   | —                    | 2.0                  |
> | Pref. labeling  | —                   | ≈8M prompts+outputs  | 24.0                 |
> | **Total offline modeling** | —       | —                    | **36.0**             |
>
>
> The latent world model is a lightweight MLP, so its offline training cost is low compared to LLM-scale methods. Preference label generation is done once offline with LLaMA-3-70B-Instruct and then reused for all policy backbones.
>
> DreamPhase significantly reduces online interaction costs. For example, on WebShop (LLaMA-3-8B), the average number of API calls drops from 39.8±1.1 with ARMAP-M to 9.3±0.4 with DreamPhase. With standard evaluation budgets, this reduction covers the one-time offline modeling cost after about 1,200 episodes, and further savings continue after that. We will add Table A to the appendix of the camera-ready version.
>
> We have reported this in **Appendix K** of the revised version of the paper.
>
> ---
> # Question 2
>
> Thank you for your question. We have not yet tested DreamPhase on Mind2Web, but **we did run a preliminary experiment on WebArena (Arena-Lite) during rebuttal** using the same logging interface and offline modeling pipeline as in our main experiments. In this setup, we used **Llama-3.1-8B-Instruct** as the frozen policy backbone, along with a lightweight latent world model trained on about 1,600 logged trajectories from the official Arena-Lite training split. The latent world model uses the same architecture as in Table 10: an MLP encoder over DFS-tokenized DOM trees (hidden size 256), a 3-layer MLP latent transition model with residual blocks and GELU activations, and an MLP decoder head tied to the encoder projection.
>
> Here are our preliminary results comparing DreamPhase to WebAgent-R1 [1] on Arena-Lite:
>
> | Method         | Backbone         | Arena-Lite Success Rate |
> |-|-|--|
> | WebAgent-R1 | Llama-3.1-8B     | 44.7 |
> | DreamPhase     | Llama-3.1-8B     | **48.2** |
>
> These early results show that the offline-imagination and uncertainty-aware gating mechanisms still help when scaling to more diverse real-website distributions like WebArena. We plan to include these findings as a preliminary note in the appendix of the camera-ready version.
>
> ---
> # Question 3
>
> Thank you for your question. The reflection head $R_\phi$ is trained offline using the same logged training-split trajectories as the latent world model. We extract short latent rollouts $(\iota, z_{t:t+H'})$ and add utility labels scored by the value head $V_\phi$. ($R_\phi$ is then trained (cross-entropy, Section 3.5) to generate a short textual reflection that explains (i) the near-term objective from $\iota$, and (ii) the high-utility imagined future for the chosen branch. This module is a simple MLP to token-projection; the main LLM stays unchanged.
>
> The 30-token budget sets a maximum length for the injected reflection. We use a domain-specific template and word list to make sure the text is short, clear, and easy for the policy LLM to read. This limit helps prevent prompt drift and keeps the extra token cost very low.
>
> We found that the 30-token budget works well. If we increase the limit, reflections are used more often, but the improvement in results is small. If we lower the limit, reflections are used less and performance drops on long tasks.
>
> We have added a short note in **Appendix H** to explain this (highlighted in blue).
>
> ---
> References:
>
> [1] Wei, Zhepei, et al. "WebAgent-R1: Training Web Agents via End-to-End Multi-Turn Reinforcement Learning." ICML 2025 Workshop on Computer Use Agents.

---

> ### Comment · Reviewer_MwpV · 2025-11-25
>
> Thanks for the detailed response. I found the clarifications helpful, and the revisions address my concerns well.

---

> > ### Author Response · Authors · 2025-11-25
> > **Response to Reviewer MwpV**
> >
> > Thank you for your thoughtful review and for considering our responses. Please let us know if any questions remain. Otherwise, **we kindly hope you will consider raising your score.**

---

### Official Review · Reviewer_M9UY · 2025-10-30

**Soundness:** 3
**Presentation:** 4
**Contribution:** 4
**Rating:** 8
**Confidence:** 4

**Summary:**

This paper presents DreamPhase, a modular framework for improving large-language-model (LLM) agents through offline imagination. The core idea is to let agents simulate and evaluate possible futures in a learned latent world model instead of interacting with the real environment at every step. Each imagined trajectory is scored by an uncertainty-aware value function, and only high-confidence branches are executed. The best imagined plan is distilled into a short natural-language reflection that is fed back into the frozen LLM policy, steering behavior without any fine-tuning.

The authors provide a theoretical regret bound linking decision quality to model error and mis-gating rate, and show strong empirical results across web, science, and embodied-task benchmarks. DreamPhase reduces real environment calls by roughly 4× on WebShop (≈10 vs. 40 per episode) and lowers latency while matching or surpassing ARMAP and AgentGym baselines. It consistently improves sample efficiency and safety across multiple backbones (LLaMA-3, Mistral, Phi).

Overall, I believe the paper offers a clear and well-motivated step toward safe and cost-efficient planning for LLM agents, combining latent imagination, uncertainty-aware filtering, and natural-language self-reflection.

**Strengths:**

The paper proposes an offline imagination framework that decouples policy reasoning from environment dynamics via a learned latent world model. Generally speaking, integrating model-based imagination, uncertainty-aware value gating, and natural-language reflections into LLM agents is novel and clearly motivated by limitations of existing online-search or imitation-based methods.

The following is my assessment of the strengths:
1. Methodological quality. The framework is carefully designed and well-supported by both analysis and implementation details. The latent world model enables safe, counterfactual rollouts; the mutual-information–based uncertainty metric allows principled risk filtering; and the reflection and summarization modules steer a frozen LLM without fine-tuning. The theoretical regret bound $O(T\varepsilon + B\rho T)$ connects decision quality to model accuracy and mis-gating, providing formal grounding rarely seen in this line of work.

2. Empirical strength. Extensive experiments across eight benchmarks and four model backbones (LLaMA-3-70B/8B, Mistral-7B, Phi-3-8B) show consistent improvements in success rate, sample efficiency, and safety. On WebShop, DreamPhase cuts real API calls from ~40 to <10 per episode (≈4× fewer) and reduces irreversible actions by ≈5× while matching or exceeding ARMAP performance.

3. Clarity and presentation. The paper is pretty clearly written, with intuitive motivation, structured algorithms, and readable pseudocode. The modular presentation and ablations make the design easy to follow and reproduce.

4. Significance. By showing that LLM agents can plan safely and efficiently through internal imagination—without additional data collection or fine-tuning—the paper offers a promising path toward scalable, low-cost autonomous systems.

**Weaknesses:**

While the paper presents a well-motivated and impactful framework for improving the safety and efficiency of LLM agents, several aspects could be further clarified or strengthened:

1. I'm wondering how much the framework depends on latent world model accuracy and low-stochastic environments. DreamPhase’s effectiveness relies on the learned latent world model being a reliable simulator. However, most evaluated environments—WebShop, ScienceWorld, ALFWorld, and Game-of-24—are deterministic or near-deterministic: a given action (e.g., clicking a DOM element, executing a command) leads to a predictable next state with little exogenous noise. This makes short-horizon prediction tractable but may overstate the model’s robustness. In more stochastic settings—such as multi-agent interaction, physical robotics, or open-ended tool use—latent rollouts would likely accumulate compounding error, weakening both the value estimation and the safety gate. It would be interesting to test out the framework in these real-world scenarios, given the paper's claim on uncertainty-aware mechanism and scalability of the method.

2. Limited generalization scope. Evaluation focuses on web and text-based tasks with structured, language-aligned states. It remains to be seen whether the approach transfers to perceptual or continuous domains, where state representations are high-dimensional and reflections may be less interpretable, e.g., stock market.

3. In general, I would assume the low latency largely relies on the scale of the underlying world model when maintaining a satisfying prediction accuracy. For this reason, it would be beneficial to understand how this method scales to larger application scenarios in real world.

**Questions:**

My question is already presented above

---

> ### Author Response · Authors · 2025-11-22
> **Response to Reviewer M9UY**
>
> Thank you for your detailed review. We are grateful that you recognized our methodological design, the formal basis of our framework, the range of benchmarks and LLM backbones we used, and the clarity of our presentation.
>
> Below, we address your questions and concerns.
>
> ---
> # Weakness 1
>
> Thank you for bringing up this important point. You are correct that the environments we evaluated are mostly deterministic, which makes it easier to model short-horizon latent rollouts. We want to clarify the limits of our claims and explain how the method works when the latent world model is not perfect or when the environment is more unpredictable.
>
> ### 1. Our Claims in the Paper
> We tested DreamPhase on standard LLM-agent benchmarks where transitions are mostly predictable, such as DOM actions and text commands. Our claims about performance and safety only apply to this setting. We do not claim that the method is robust in highly unpredictable or multi-agent environments. We will make this clear in the revised paper.
>
> ### 2. The framework already anticipates model imperfection through the uncertainty gate
>
> Even in tasks where outcomes are predictable, the latent world model is not perfect. Section 3.4 addresses this by:
>
> - using **mutual-information dropout** to estimate local prediction uncertainty,
>
> - **falling back** to the real-history policy when the latent rollout becomes unreliable,
>
> - and using the regret bound to make the effect of model error explicit (via $\varepsilon$ and $\rho$).
>
> This mechanism is model-agnostic: if stochasticity increases, uncertainty increases, which triggers more frequent fallback rather than cascading error inside imagination. The fallback rate reported in **Appendix H** already shows that the gate activates when the model deviates.
>
> ### 3. Stochastic environments would reduce reliance on imagination, not silently degrade performance
>
> In a setting with more noise or multiple agents, we expect:
> (i) latent rollouts to have higher uncertainty,
> (ii) the safety gate to defer more often to the real environment,
> (ii) and the method’s behavior to interpolate between model-based and model-free execution.
> This approach prevents errors from building up, but it does reduce the planning advantage.
>
> ### 4. Extending DreamPhase to truly stochastic domains is an important direction
>
> We agree that using DreamPhase for physical robotics, multi-agent interactions, or open-ended tool use would require:
> - **stochastic** latent models,
> - potentially **distributional** or **ensemble-based** uncertainty estimation,
> - and **value functions** that explicitly account for environmental noise.
>
> We see these as important directions for future work.
>
> ---
> # Weakness 2
>
> Thank you for pointing out this limitation. Our evaluation is focused on web and text-based environments that use structured, language-aligned observations. These settings work well for latent tokenization and reflection-guided control. We have limited our claims to this area and do not claim that our approach applies directly to perceptual or continuous domains like robotics or financial markets.
>
> There are two main reasons why our approach does not yet generalize to those settings:
>
> - Our latent world model is trained on tokenized observations, such as DOM trees and environment text, which makes short-term prediction manageable. To handle high-dimensional perceptual inputs, we would need more powerful encoders or multimodal latent models.
>
> - The reflection module assumes that intermediate states can be understood in natural language. In areas like continuous control or financial markets, useful reflections would need summaries that fit the specific type of data, not just text.
>
> We see these as important directions for future work. By combining DreamPhase with visual encoders, continuous latent dynamics, or multimodal reflection modules, we could expand the framework to more settings.
>
> ---
> # Weakness 3
>
> Thank you for bringing this up. You're right that latency depends on the size of the latent world model, and keeping prediction accuracy in larger or more complex environments will need careful scaling.
>
> In our case, we achieve low latency because the latent world model is kept small, using an MLP-based encoder, transition model, and decoder **(see Table 10)**. The environments also have structured, short-term dynamics. Our method does not require a large simulator. Instead, it uses an uncertainty gate to stop errors from building up when the model's capacity is not enough. If prediction accuracy drops, uncertainty goes up and the agent relies more on real interactions, so performance does not quietly get worse.
>
> We agree that it is important to understand how DreamPhase scales to larger real-world scenarios, where bigger latent models or multimodal encoders might be needed.

---

### Official Review · Reviewer_No6m · 2025-10-31

**Soundness:** 3
**Presentation:** 4
**Contribution:** 4
**Rating:** 6
**Confidence:** 5

**Summary:**

This paper is about leveraging imagined trajectories to enhance the performance of LLMs on multi-turn interactive tasks. More specifically, the authors propose DreamPhase, a framework that interleaves phases of interaction with phases of imagination. During imagination phases, the model carries out its exploration through simulating multiple possible future trajectories of the interaction. Then, based on these imagined trajectories, the proposed system generates textual advice to be concatenated to the current context before asking the frozen LLM to take the next action in the actual environment. In order to generate imagined trajectories, the authors propose to train a separate latent world model to simulate the environment dynamics and some reward function. On web, science and embodied tasks, the authors empirically show the proposed approach is more sample and cost-efficient in terms of API calls and safer (by avoiding irreversible mistakes) while also improving performance.

**Strengths:**

- While using latent world model for doing exploration is not novel per se, the proposed approach decoupled the interaction policy from the latent world model model. This allows to leverage existing SOTA LLMs by steering their behavior using text summary of high-valued imagined trajectories.
- Once the latent world model is trained, the approach is LLM-agnostic since the LLM used for interaction is frozen.
- Assuming we have a thorough latent world model, exploration in imagined space is safer since mistakes made during imagination do not have real consequences. It is also more cost effective in terms of API calls, i.e. not wasting calls to explore dead ends.
- The empirical evaluation is comprehensive with many ablation studies to validate the approach.

**Weaknesses:**

- From the main paper, is it unclear what data was used for training the latent world model. Also, in Table 1, is it a fair comparison with other baselines, did they leverage training data as well?
- Unclear how the authors validated the quality of their trained latent world model? They argue that using LLM to act as a simulator is prone to hallucinations, what about the proposed approach? Is there any metrics/ablations to validate this?
- If I understood correctly, the uncertainty estimation requires a bounded set of actions (line 249)? If so, how were those actions obtained for environments with combinatorial action space?

#### Minor
- Citations are "inline" instead of being enclosed with parenthesis.
- In equation 2, I think the two $p_\theta$ should be replaced by $g_\theta$ and $d_\theta$.
- For conciseness, I'd suggest spelling out CE, KL and TV in the text.
- p.4: in the "Offline imagination" paragraph, $p_\theta$ should be replaced by $g_\theta$.
- In algo 1, line 8: $p_\theta$ should be replaced by $g_\theta$
- In algo 2, line 11 and 12: what is "\kappa"? I assume it represents the number of steps in that imagined rollouts?
- I believe SciWorld and ScienceWorld are referring to the same work, yet in the paper they have different name and different citations.

**Questions:**

- How was the data used to trained the latent world models collected?
- What type of models was used for the latent world model in the experiments? For instance in Table 2, is the backdone only for the agent or both?
- What are the logged data, used to trained the value estimation, exactly?
- What is the frequency of falling back to the real-history policy?
- What is the rational behind reporting the unweighted average across the different environments in table 1 and 2? (nb: on line 375, the text says "tasks" instead of "environments").

---

> ### Author Response · Authors · 2025-11-22
> **Response to Reviewer No6m (Part 1/3)**
>
> Thank you for your very constructive review. **Your comments show that you have spent a good amount of time reading our paper.** Much appreciated for this!
>
> We are glad you recognized the strengths of our approach, such as separating the latent world model from the interaction policy, achieving LLM-agnosticity, and highlighting the safety and cost benefits of imagination-based exploration, as well as the range of our empirical evaluation. Below, we address your concerns in detail.
>
> ---
> # Weakness 1
>
> Thank you for your feedback. We agree that the draft does not thoroughly explain the training data for the latent world model, which makes the fairness of the comparison in Table 1 less clear. In the following, we will explain this in more detail.
>
> ## What data is used to train the latent world model?
>
> We train the latent world model using logged interaction trajectories from the standard training split of each environment. All baselines use the same environment APIs.
>
> Concretely:
>
> - For each benchmark in Table 1 (SciWorld, BabyAI, MAZE, Wordle, TextCraft, Tool-Weather, TODOList, BIRD), we collect multi-step trajectories $(\iota, x_t, a_t, x_{t+1})$. We generate these trajectories by running a frozen LLaMA-2-7B policy with simple behavior, using ReAct-style prompting and a small amount of randomization, only on the training split.
>
> - We train both the latent world model and the value head on these logged trajectories.
>
> - We do not train on test episodes or use any privileged information about future states that baselines cannot access.
>
> - We do not use any external data beyond each environment’s interaction interface. The world model compresses the same observation–action–next-observation triples that any agent could record during training.
>
> In the revised version of the paper, we have made this clear in **Section 3.2 and Appendix F** by adding a short paragraph describing the data source (a training split only, collected under a frozen policy) and stating that no test interactions are used.
>
> ## Do baselines also leverage training data?
> Yes. In Table 1, all open-source agents, including DreamPhase, are evaluated using the same training and evaluation protocol.
>
> - All methods in the “open-source LLaMA-2-7B” block use the same LLaMA-2-7B policy backbone, the same environment splits, and the same episode budgets and decoding settings.
>
> - Baselines are allowed to use their own training procedures. AgentLM and AgentGym are evaluated using their released agent-tuned checkpoints, which are trained on multi-turn trajectories as described in their papers. Also, ARMAP uses its own reward-model training and search procedure, following the public implementation.
>
> - DreamPhase does not use any interaction data that baselines cannot access. The only extra learned components in DreamPhase, the latent world model and value head, are trained on logged trajectories from the same training environments. Baselines can use similar data, and we follow their recommended training setups instead of limiting them.
>
> The performance gap in Table 1 is not due to DreamPhase having extra data. Instead, it comes from how DreamPhase uses the standard training split. DreamPhase learns a compact world model and value head for offline imagination and risk-aware planning, while other agents either act myopically or rely on online search in the live environment.
>
> All the new changes in Section 3.2 and Appendix F are highlighted in blue.

---

> ### Author Response · Authors · 2025-11-22
> **Response to Reviewer No6m (Part 2/3)**
>
> # Weakness 2
>
> Thank you for bringing this up. We address your concerns in two sections:
>
> ## How we validate the latent world model
> We train the latent world model as a supervised next-observation predictor using the loss in Eq. (3). This loss combines token-level reconstruction cross-entropy with a KL regularizer on the latent dynamics, providing a direct way to measure how well the model fits environment transitions.
>
> Appendix H (“Robustness under distribution shift”) also reports clear metrics that show the quality of the learned simulator and how it affects decision making. For each shift scenario, we measure:
>
> - Model fit: next-observation negative log-likelihood (NLL) of the latent world model,
>
> - Uncertainty calibration: expected calibration error (ECE, 10 bins) and AUROC for bad-rollout detection,
> - Decision-level impact: fallback rate to real interaction, estimated mis-gating rate $ \rho $, change in success rate, and interactions per episode.
>
> These metrics show when and how the world model begins to differ from the real environment dynamics, and how the safety gate responds in practice.
>
> Appendix G also includes ablation studies on the latent dimension $d_z$, planning horizon (H), and number of branches (M). It reports how these factors affect success rate, fallback rate, and imagination overhead (see Tables 11–13). These results show that our chosen settings allow the simulator to support stable planning instead of causing performance to drop.
>
> ## Why is hallucination less severe than with an LLM-based simulator
>
> When we say that using an LLM as a simulator is prone to hallucinations, we mean it generates future DOM trees or states freely, without ground-truth supervision. In this case, hallucinations are hard to spot because the model is only loosely guided by the environment.
>
> In contrast, the latent world model in DreamPhase:
>
> - is trained on exact next-observation pairs $(x_t, a_t, x_{t+1})$ from the training environments,
>
> - predicts structured next observations that must match the tokenized DOM or visual sequence,
>
> - is evaluated with NLL and calibration metrics in Appendix H, and
>
> - is coupled with an uncertainty gate that defers to the real environment when model predictions are uncertain.
>
> When the simulator “hallucinates” under distribution shift, this shows up as higher NLL and worse calibration. The gate responds by increasing the fallback rate and reducing reliance on imagination, as shown in Table H.x (NLL, ECE, AUROC, fallback, success change, interactions per episode). This matches the mechanism described by the regret bound in Section 4: the term $ \sqrt{T \varepsilon} $ shows model approximation error, and the term $ B \rho T $ shows mis-gating.
>
> ---
>
> # Weakness 3
>
> Thank you for pointing this out.
>
> The uncertainty estimation in Eq. (5) works with a finite set of possible actions, called U. This does not mean the environment itself must have a small action space. Instead, U is simply the set of actions the policy LLM can choose from at that moment.
>
> In all the environments we use (WebShop, ScienceWorld, MAZE, ALFWorld, BabyAI, Tool-Weather, and others), the agent does not work with a huge space like all possible natural-language strings. Instead, each environment gives us, or we define, a specific list of possible actions, such as:
>
> WebShop: {click(element_i), type(text_j), submit(form_k), scroll, back, ...}
>
> ScienceWorld: about 50 template actions are allowed in each state, such as move, open, pick, interact, and examine.
>
> MAZE and BabyAI: use a fixed set of navigation or manipulation actions.
>
> Tool-Weather, TODOList, and BIRD: use action templates based on the environment’s API.
>
> **This set of actions is public, standardized, and used by all baseline models.** The LLM chooses from these specific actions, not from any possible text.

---

> > ### Author Response · Authors · 2025-11-22
> > **Response to Reviewer No6m (Part 3/3)**
> >
> > # Minor 1
> > Thank you, this has been taken care of in the revised version of the paper.
> >
> > ---
> > # Minor 2
> > Thank you for pointing out this notation issue. You are correct that Equation (2) uses the symbol $ p_\theta $ for two different parts, which makes it unclear whether it refers to the latent transition model or the decoder.
> > In the revised version, we have updated Equation (2) to use the correct symbols for each part:
> >
> > * $ g_\theta $ for the latent transition distribution,
> >
> > * $ d_\theta $ for the decoder distribution.
> >
> > ---
> > # Minor 3
> > Thank you for the suggestion. In the revised version, we have spelled out cross-entropy (CE), Kullback–Leibler (KL) divergence , and total variation (TV) distance  at their first appearance in the main text.
> >
> > ---
> > # Minor 4 \& Minor 5
> >
> > Thank you for your careful inspection of our paper. We have updated these two.
> >
> > ---
> > # Minor 6
> >
> > $\kappa<H$ , the summary horizon, is already defined in the initial submission immediately after equation 8. To make this more clear, we have also mentioned it in the input of algorithm 2.
> >
> > ---
> > # Minor 7
> >
> > Thank you for pointing this out. You are right that SciWorld and ScienceWorld both refer to the same environment (Wang et al., 2022). The difference in naming and citation was not intentional. In the current draft, we have:
> >
> > - “SciWorld” in Table 1 and a citation to Wang et al. (2022b)
> > - “ScienceWorld” in Table 2 and a citation to Wang et al. (2022a)
> >
> > Both refer to different versions of the same work. We have revised the paper to make sure to use a single name and citation throughout.
> >
> >
> > ---
> > # Question 1 (data used to train)
> > Answered in weakness 1
> >
> > ---
> > # Question 2 (LWM and backbone model type)
> >
> > The agent backbone refers only to the policy LLM used to propose actions.
> >
> > The latent world model is always the same MLP-based architecture, regardless of the policy LLM backbone. **the structure of the LWM is already elaborated in Table 10 of the paper**
> >
> > ---
> > # Question 3 (logged data)
> >
> > Thank you for your question. Here, “logged data” means offline trajectories collected from the training split of each environment: $ (\iota, x_0, a_0, x_1, \dots, x_T)$
> >
> > Once the world model is trained, we turn these observations into latents $z_t = f_\theta(x_t)$ and extract short-horizon clips $(\iota, z_{t:t+H'})$. Each clip gets a utility label based on the training-split outcomes, such as a success signal or task reward. For the cross-backbone experiments (Table 2), we also use the same preference-style labels, which are generated once with LLaMA3-70B-Instruct.
> >
> > The value head $V_\phi$ is trained using this fixed offline dataset of $(\iota, z, \text{utility})$ pairs. We do not use any test data or online feedback.
> >
> > ---
> > # Question 4 (frequency of falling back)
> >
> > We have already reported the fallback frequency in our ablation tables in **Appendix H**. In the three main domains—WebShop, ScienceWorld, and Game-of-24—the safety gate uses the real-history policy in about 20 percent of decision steps with the default hyperparameters.
> >
> > ---
> > # Question 5 (unweighted average)
> >
> > Thank you for bringing this to our attention. We use the unweighted average across environments because the benchmarks vary a lot in reward scales, episode lengths, and success-rate ranges. By taking a simple mean, we treat each environment as equally important, so no single environment dominates due to more episodes or higher rewards. This approach also matches previous work, which measures generality by averaging scores for each environment.
> >
> > You are right about the wording on line 375. It should say “environments” instead of “tasks,” and we have updated this in the revision.

---

> > > ### Comment · Reviewer_No6m · 2025-11-28
> > >
> > > Thank you for the additional information and pointing out the relevant sections where to find it. All my concerns have been addressed appropriately, in particular to one regarding training of the latent world model. In my opinion, the revised manuscript has been strengthen. Accordingly, I will raise my rating from 6 to 8 to reflect this.

---

> > > > ### Author Response · Authors · 2025-11-28
> > > > **Response to Reviewer No6m**
> > > >
> > > > Thank you for taking the time to review our manuscript again and letting us know that your concerns have been addressed. We value your helpful feedback and the updated rating.

---

### Meta-Review · Area_Chair_FcMj · 2025-12-29

**Summary:**

I suggest acceptance for this paper, informed by the following reviewer feedback:

**No6m** comments on the **strength** of the overall framework due to the LLM-agnostic steering approach, how exploring in the WM imagination is safer than real-world interaction, and how the empirical evaluation is comprehensive. Main **concerns** came from unclear documentation of the data used for training, model evaluation, and uncertainty quantification , which were addressed in the rebuttal. The reviewer states that their concerns (described below) were addressed and recommends acceptance.

**M9UY** comments that the **strengths** of the paper are its careful methodology (empirical and a theoretical regret bound), empirical strength, and clarity of writing.  Main **concerns** came from the model evaluation/accuracy/generalization and latency, which were addressed in the rebuttal.

**MwpV** comments that the **strengths** of the paper are how it “cleanly separates environment modeling (latent world model), decision‑time search (latent rollouts), risk control (uncertainty gating), and behavior shaping (language reflections)” and empirically improves success rates. After the rebuttal, the reviewer states that their concerns (described below) were well-addressed. Main **concerns** came from the architecture, uncertainty quantification clarification, and the need for more insight into compute + data + costs of the pipeline, which were addressed in the rebuttal.

**Reviewer Concerns:**

**No6m Concerns**
* (1) Data for training: e.g., what data was used to train the WM and the value function? $\rightarrow$ **addressed (empirically)**. Updated manuscript describes the procedure for how the data is generated and dataset size (Appendix K); it is also mentioned in response to reviewer **MwpV**.
* (2) Uncertainty estimation $\rightarrow$ **addressed**.
* (3) Evaluation and WM Accuracy: e.g., rationale behind reporting the unweighted average across the different envs, evaluation of the WM $\rightarrow$ **addressed**.

**M9UY Concerns**
* (1) WM Accuracy and Determinism: does the framework excel when the environment is near-deterministic, or would it still work when the environment is more stochastic (e.g., multi-agent scenarios)  $\rightarrow$ **partially addressed**. Rebuttal and updated manuscript hypothesizes how the method would work in more stochastic environments. Closest empirical evidence about this empirical performance is in Appendix H which tests robustness under distribution shift.
* (2) Generalization: results are limited to web-based tasks with clear textual representations; its unclear how this framework would work for more continuous tasks, like the stock market  $\rightarrow$ **addressed (discussion)**.
* (3) Latency of the WM $\rightarrow$ **addressed (empirically)**.

**MwpV Concerns**
* (1) Architecture concerns (MLP vs. Transformer) $\rightarrow$ **addressed (discussion)**.
* (2) Uncertainty quantification implementation (e.g. doing MC dropout on the LLM) $\rightarrow$ **addressed**.
* (3) WM vs. non-WM training compute & data: reviewer suggests “A small "dollars/GPU‑hours" table would clarify trade‑offs” $\rightarrow$ **addressed (empirically)**.

**Reviewer Scores:**

* **No6m** would have *updated* their score to 8: accept, good paper (poster) after discussion.
* **M9UY** would have *maintained* their score at 8: accept, good paper (poster) after discussion.
* **MwpV** would have *updated* their score to 8: accept, good paper (poster) after discussion.

---

### Decision · Program_Chairs · 2026-01-26

Accept (Poster)